# Stranded short nascent strand sequencing reveals the topology of DNA replication origins in *Trypanosoma brucei*

**Slavica Stanojcic[1]\*[†], Bridlin Barckmann[1][†], Pieter Monsieurs[2], Lucien Crobu[1], Simon George[3], Yvon Sterkers[1]\***

[1]University of Montpellier, CNRS, IRD, Academic Hospital (CHU) of Montpellier, MiVEGEC, Montpellier, France; [2]Trypanosoma unit, Department of Biomedical Sciences, Institute of Tropical Medicine, Antwerp, Belgium; [3]MGX-Montpellier GenomiX, University of Montpellier, CNRS, INSERM, Montpellier, France

**\*For correspondence:**
slavica.stanojcic@umontpellier.fr (SS);
yvon.sterkers@umontpellier.fr (YS)

[†]These authors contributed equally to this work

**Competing interest:** The authors declare that no competing interests exist.

## eLife Assessment

The authors adapt sequencing of nascent DNA (DNA linked to an RNA primer, "SNS-Seq") to map DNA replication origins in *Trypanosoma brucei*. The main impact of this work is reporting a new set of putative origins, which do not overlap with previously reported origins, but which appear to overlap with previously mapped DNA-RNA hybrid (R-loops). Thus, these **valuable** findings open up new avenues for further investigation into the mechanistic basis for firing of replication forks in this organism. However, the supporting evidence remains **incomplete** and would benefit from orthogonal validation. This work will be of interest to those studying DNA replication and epigenetic regulation of fork origins.

**Abstract** The universal features that define genomic regions acting as replication origins remain unclear. In this study, we mapped a set of origins in *Trypanosoma brucei* using stranded short nascent strand sequencing methods. Our results showed that DNA replication predominantly initiates in intergenic regions between poly(dA)- and poly(dT)-enriched sequences. G4 structures were detected in the vicinity of some origins and were embedded in poly(dA)-enriched sequences in a strand-specific manner: G4s on the plus strand were located upstream while those on the minus strand were located downstream of the centre. The origins' centres were found to be areas of low nucleosome occupancy, surrounded by regions of high nucleosome occupancy. Furthermore, our results demonstrate that 90% of replication origins overlap with a minor proportion of the previously reported RNA: DNA hybrids. These findings shed new light on the sequence and structural features that define the topology of replication origins in *T. brucei*. To further characterise replication dynamics at the single-molecule level, we employed DNA combing analysis.

## Introduction

DNA replication is one of the most fundamental processes by which a cell creates an exact copy of its genome prior to division. In eukaryotic cells, DNA replication initiates at multiple genomic sites, which are known as origins. Despite extensive research, a comprehensive universal code that is common for all eukaryotic origins remains elusive (reviewed in *Lee et al., 2023*; *Hulke et al., 2020*; *Leonard and Méchali, 2013*). The first isolated and characterised eukaryotic origin comes from budding yeast, *Saccharomyces cerevisiae*, where DNA replication starts from the AT-rich autonomously replicating sequence (ARS) with a loosely defined and thymidine-rich ARS consensus sequence (ACS; *Newlon*

*and Theis, 1993*). The other eukaryotic origins lack a unique consensus sequence. Some studies reported increased AT-content (*Okuno et al., 1999*; *Kong and DePamphilis, 2001*; *MacAlpine et al., 2004*; *Balasov et al., 2007*; *Stanojcic et al., 2008*; *Segurado et al., 2003*; *Mojardín et al., 2013*) or long poly(dA:dT) tracts as replication origins (*Eaton et al., 2010*; *Tubbs et al., 2018*). Other reports described the origins as regions of a high GC-content, such as CpG islands (*Delgado et al., 1998*; *Cadoret et al., 2008*; *Cayrou et al., 2011*; *Pratto et al., 2021*) or regions near G-rich sequences with the potential to form G-quadruplexes (G4s; *Besnard et al., 2012*; *Valton et al., 2014*; *Cayrou et al., 2015*; *Comoglio et al., 2015*; *Lombraña et al., 2016*; *Langley et al., 2016*; *Sequeira-Mendes et al., 2019*; *Akerman et al., 2020*; *Prorok et al., 2019*). In addition, several epigenetic factors and chromatin environments were linked to eukaryotic origins, but without a common signature (*Lee et al., 2023*; *Lombraña et al., 2016*; *de O Vitarelli et al., 2024*). However, replication origins exhibit several characteristics that distinguish them from the surrounding chromatin. They appear in nucleosome-depleted regions (*Lipford and Bell, 2001*; *Berbenetz et al., 2010*; *Foss et al., 2024*; *Poulet-Benedetti et al., 2023*), which are flanked on one or both sides by well-positioned nucleosomes (*Eaton et al., 2010*; *Berbenetz et al., 2010*; *Foss et al., 2024*; *Poulet-Benedetti et al., 2023*; *Jansen and Verstrepen, 2011*). An active origin is characterised by the generation of two divergent single-stranded DNA molecules, each with a short RNA primer at the 5' end, known as short nascent strands (SNS; reviewed in *Nasheuer and Meaney, 2024*). Size-selected SNS can be enriched and purified from DNA molecules by treatment with T4 polynucleotide kinase (T4 PNK) and $\lambda$-exonuclease, and this approach was developed for mapping of DNA replication initiation sites in yeast (*Bielinsky and Gerbi, 1998*). Thereafter, this technique has been successfully applied in different species, using approaches such as SNS-on-chip (*Cadoret et al., 2008*; *Cayrou et al., 2011*) or by high-throughput sequencing of SNS (SNS-seq; *Besnard et al., 2012*; *Cayrou et al., 2015*; *Lombraña et al., 2016*; *Akerman et al., 2020*; *Prorok et al., 2019*; *Foulk et al., 2015*; *Castellano et al., 2024*). A recent development has improved the conventional SNS-seq method by enabling the preservation of SNS molecule directionality following sequencing (*Pratto et al., 2021*; *Senkevich et al., 2015*).

*Trypanosoma brucei* is a unicellular parasite of medical and veterinary importance that provokes deadly vector-borne diseases: sleeping sickness in humans and nagana disease in livestock. This parasite circulates between mammalian and insect hosts, developing various life cycle stages: bloodstream form (BSF) in the mammals and the procyclic form (PCF) in the midgut of the tsetse fly (*Fenn and Matthews, 2007*). The order *Trypanosomatida* diverged independently of the *Opisthokonta*, the group comprising animals and fungi (*Baldauf, 2003*), and has attracted attention as an original model for biological and comparative studies. These single-celled eukaryotes have some unusual genomic and epigenetic properties. Transcription is polycistronic, meaning that multiple functionally unrelated genes are transcribed into one long precursor mRNA (*Clayton, 2019*). A better understanding of the fundamental processes in these divergent eukaryotes could provide insight into evolutionary steps and pave the way for innovative medical treatments for sleeping sickness and nagana disease.

The replication of DNA in these parasites has recently become a subject of interest (*Stanojcic et al., 2016*; *Tiengwe et al., 2012*; *Devlin et al., 2016*). To map and identify active DNA replication origins (hereafter referred to as 'origins') in *T. brucei*, we isolated RNA-primed SNS molecules and sequenced them using a stranded protocol that preserves their orientation. This stranded SNS-seq method allowed the mapping of origins between two divergently oriented SNS peaks and the discovery of potential DNA replication start sites. We then conducted an extensive bioinformatics analysis of the mapped origins and compared our findings with previously published scientific datasets. This enabled us to identify several key components of origins and to propose a model of an origin in *T. brucei*.

## Results
### Origin mapping by stranded SNS-seq

Stranded SNS-seq method was employed to map the origins in two distinct cell types of *T. brucei*, representing two life cycle stages of this parasite: PCF and BSF cells. The experimental workflow for purifying and enriching SNS, together with the corresponding control, is illustrated in *Figure 1A*. SNS-enriched samples were prepared through a two-stage process. SNS molecules were enriched first by size-selection (0.5–2.5 kb) and second, by removing broken DNA molecules by $\lambda$-exonuclease

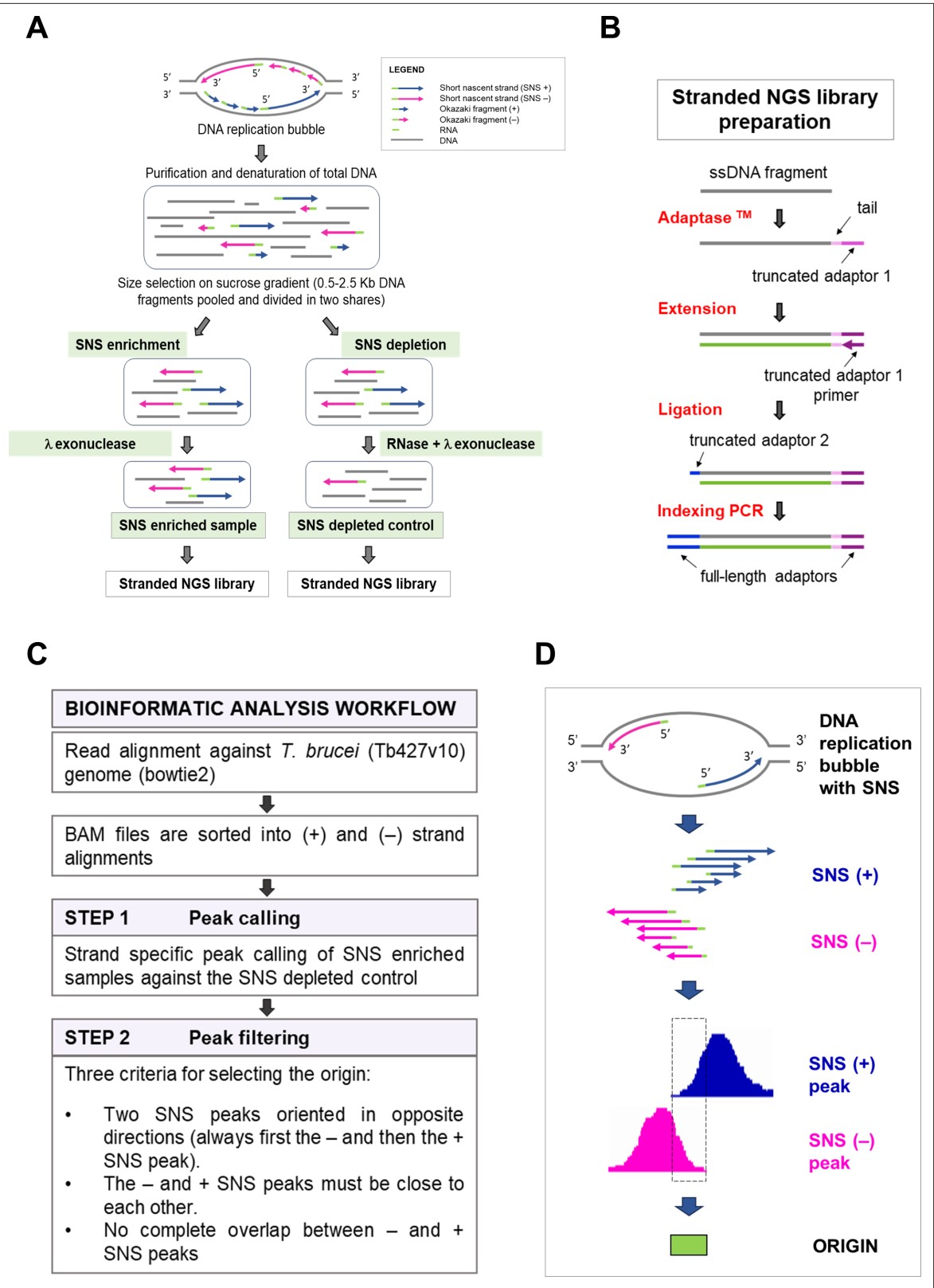

**Figure 1.** Experimental protocol and bioinformatics workflow for selecting the origins. (**A**) Schematic representation of the experimental workflow for short nascent strand (SNS) purification. A replication bubble with DNA replication intermediates and their orientation is shown at the top. The methodology employed for the generation of the SNS-enriched samples and corresponding SNS-depleted controls is illustrated. (**B**) Workflow of stranded library preparation for Illumina sequencing. Single-stranded DNA fragments were tailed and tagged at the 3' end with truncated adaptor 1.

*Figure 1 continued on next page*

*Figure 1 continued*

Primer extension was used to generate the complementary strand. The ligation step was used to add the truncated adaptor 2. The indexing PCR step included full-length adaptors. (**C**) Schematic diagram of the bioinformatic workflow applied for stranded SNS-seq data analysis. Two steps of peak selection were performed to map the origins. The first step was peak calling against SNS depleted control. The second step was peak filtering, based on the three indicated criteria. (**D**) The schematic representation shows one origin, generating a replication bubble with two SNS leading strands in divergent orientation (SNS +and SNS –). The plus and minus SNSs are presented to illustrate the growth of the leading strand in both directions after the ligation of the upstream Okazaki fragments to the 5' ends of SNS. The different sizes of the isolated SNS are also presented. The pink peak represents the SNS minus peak synthesised on the plus DNA strand. The blue peak represents the SNS plus peak synthesised on the minus DNA strand. The dashed rectangle, situated between the two inner borders of two divergent SNS peaks, and the green box represent the mapped origin.

treatments. SNS-depleted control, prepared in parallel with every SNS-enriched sample, underwent an additional RNase treatment before $\lambda$-exonuclease treatment to remove RNA, including RNA primers from the 5' end of the SNS (Methods and *Figure 1A*). SNS-depleted control contained molecules resistant to RNase and/or $\lambda$-exonuclease treatment, representing all molecules that are difficult to digest with lambda exonuclease. This control was utilised as a background control. Then, stranded libraries were prepared for high-throughput sequencing (*Figure 1B* and Methods). Paired-end sequencing of SNS-enriched samples and SNS-depleted controls was performed on three biological replicates of PCF and BSF cells. Read counts after sequencing and each analysis step are shown for each sample in *Supplementary file 1* and *Figure 2—figure supplement 1A*. The bioinformatics workflow used for origin selection is schematically presented in *Figure 1C* and Methods. In brief, after the alignment and strand-specific sorting of the reads, two steps of peak selection were performed. The first step was strand-specific peak calling against SNS-depleted control as a background control. The aim of this step was to eliminate false positive peaks resulting from undigested DNA structures, such as G4s and other structures that could withstand $\lambda$-exonuclease treatment. The second step, called 'peak filtering', identified peak pairs consistent with the divergent orientation of SNS molecules at active origins (*Figure 1C*). Therefore, only peak pairs with a minus peak in front of a plus peak and both in proximity (Methods and *Figure 1C and D*) were retained as positives, while 88–96% of the called peaks were removed in the 'peak filtering' step (*Figure 2—figure supplement 1B and C*). Here, we used information about the divergent orientation of SNS, a key feature of active origins, to increase the specificity of origin mapping. Ultimately, the origins were identified as genomic regions situated between the inner borders of filtered peak pairs (*Figure 1D*).

The stranded SNS-seq data were mapped across the entire genome, including 11 core regions and all non-core regions (subtelomeric and 'unitig') (*Müller et al., 2018*). A representative screenshot of the distribution of plus and minus peaks and the positions of the mapped origins for three replicates of BSF cells is shown in *Figure 2A*. We found a variable number of origins in the replicates (*Figure 2—figure supplement 1D*), and the overlap of origins between replicates varied from 8–64%. The distribution of the lengths of the mapped origins was estimated for each replicate (*Figure 2—figure supplement 2A*), and the average length of origins was found to be approximately 150 bp. We are most likely affecting the average length by applying filtering and setting the maximum distance between the positive and negative peaks, which excludes potentially wider origins. After mapping, we merged the origins from the three replicates, identifying a total of 914 origins in BSF cells and 549 in PCF cells. Of the origins in BSF cells, 742 (81.2%) were mapped to the core genomic region and 172 (18.8%) to the non-core regions. In PCF cells, 492 origins (89.6%) were mapped to the core genomic region and 56 (10.4%) to the non-core regions (*Figure 2B*). Therefore, although these regions present more than half of the genomic sequence (*Figure 2B*), we mapped fewer origins in non-core regions.

To identify preferred genomic sites of origin activation, we analysed origin distribution around centred intergenic and genic regions. Origins were unevenly distributed in the genome, showing higher density in intergenic and lower in genic regions (*Figure 2C*). Within intergenic regions, origins were evenly distributed. The intergenic enrichment was consistent across both cell types and absent in shuffled controls, which consist of randomly selected regions matched for size, number, and chromosomal distribution (*Figure 2C*). Subsequently, the proportion of origins in four distinct genomic regions was determined, revealing that over 92% of PCF and 85% of BSF origins map within intergenic regions (*Figure 2D*). This distribution was significantly different from the distribution that would be expected if the origins were distributed evenly across the genome (p-value <0.0001; *Figure 2D*).

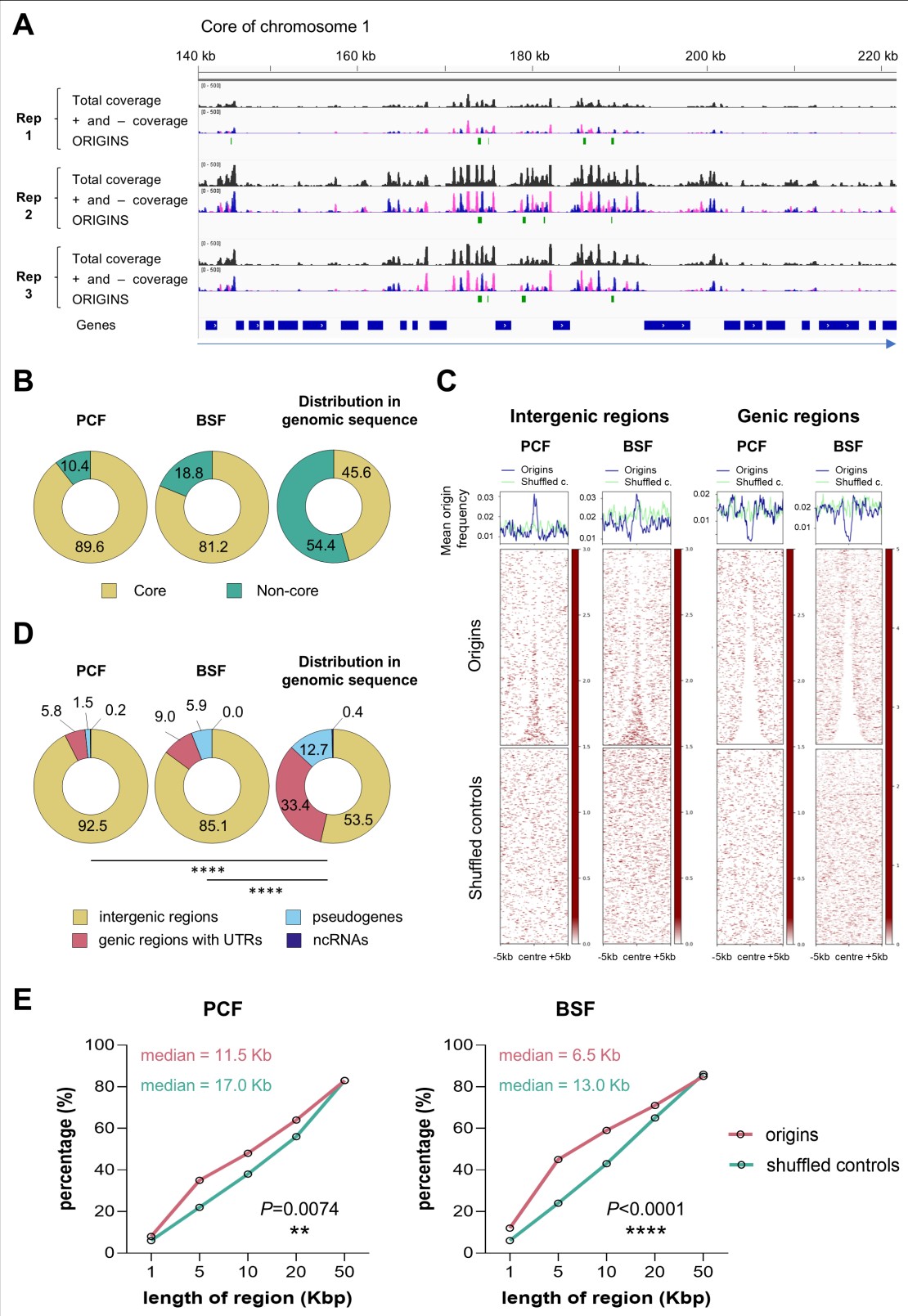

**Figure 2.** Genomic distribution of mapped origins. (**A**) A representative screenshot from Integrative Genomics Viewer (IGV) presenting the read coverage and the positions of the mapped origins in a ~80 kb long genomic region of the core chromosome 1. The three replicates of BSF cells are presented. The total read coverage is in black, the minus and plus strand read coverage are in pink and blue, respectively. The green bars represent the mapped origins. The blue bars show the position of the genes and the direction of transcription of the polycistronic unit is indicated by the blue

*Figure 2 continued on next page*

*Figure 2 continued*

arrow below. (**B**) The pie charts show the proportions of mapped origins within the core (yellow) and non-core (green) regions in PCF and BSF cells, as well as the distribution of these two regions within the genomic sequence. (**C**) Heatmaps present the distribution of the mapped origins and shuffled controls within centred genic and intergenic regions (±5 kb). Shuffled controls present random genomic regions chosen with respect to the size and chromosomal distribution of origins. (**D**) The pie charts illustrate the proportions of the mapped origins of PCF and BSF cells within four indicated genomic regions, as well as the distribution of these four regions within the genomic sequence. The p-values were computed using Chi-square tests, which involved a comparison of the absolute numbers of indicated categories for each pair of datasets (**** - p<0.0001). (**D**) Empirical cumulative distribution function (ECDF) showing the proportion of pairwise distances between origins (pink) or between shuffled control regions (green). The p-values were computed using Chi-square tests.

The online version of this article includes the following figure supplement(s) for figure 2:

**Figure supplement 1.** Quantification of read count, called and filtered peaks and detected origins.

**Figure supplement 2.** Quantitative Features of Replication Origins.

We then wanted to see if origins were organised as solitary sites or grouped together, so we examined how they clustered within genomic regions of increasing length. Our results showed that *T. brucei* origins were predominantly organised in clusters, with 85.8% of PCF origins and 83.2% of BSF origins organised in clusters of 50 kbp (*Figure 2—figure supplement 2B*). Empirical cumulative distribution function (ECDF) was calculated for both cell types, and a statistically significant difference was confirmed between the origins and shuffled controls clustering (*Figure 2E*). The results showed that 50% of the PCF origins were clustered within 11.5 kb regions (shuffled control median was 17 kb), while 50% of the BSF origins were clustered within 6.5 kb regions (shuffled control median was 13 kb; *Figure 2E*). This means that origins tend to be activated within initiation zones or clusters rather than as solitary initiation sites in *T. brucei*. This finding is consistent with previous research in other organisms, which has shown that origins are unevenly distributed and tend to form clusters of different lengths (*Cayrou et al., 2015*; *Lombraña et al., 2016*; *Akerman et al., 2020*). Next, the intersection of PCF and BSF origins revealed that 238 origins are in common (*Figure 2—figure supplement 2C*). The number of origins in both cells was compared; for this, the reads were normalised to have an equal number of mapped reads in both cell types (Methods, Read sub-sampling). We showed that BSF cells contained almost twice as many origins as PCF cells (BSF: 844 origins and PCF: 454 origins; *Figure 2—figure supplement 2D*).

We then calculated the inter-origin distances (IODs) distribution at the population level for both cell types. The calculated population-based IODs were shorter in BSF (median 15.3 kb) compared to PCF cells (median 21.4 kb) and this difference was statistically significant (*Figure 2—figure supplement 2E*). Very long IODs were rarely observed, with maximum distances of 354 and 488 kilobases for PCF and BSF origins, respectively (*Figure 2—figure supplement 2E*).

## Insights from DNA molecular combing: comparative analysis of single-molecule DNA replication and cell population origin use

Stranded SNS-seq and DNA molecular combing are two complementary techniques. Stranded SNS-seq provides a genome-wide overview of active origins within a given cell population. However, it should be noted that the usage of these origins may vary between individual cells within the same population (reviewed in *Fragkos et al., 2015*). DNA combing provides an overview of DNA replication parameters at the single-molecule level. In this study, we used DNA combing to compare the replication of PCF and BSF cells. Replicating cells were labelled with two modified nucleosides, followed by immuno-detection (Methods). After immuno-detection, the three discrete signals were discerned on combed DNA. The red signal, followed by green, indicated the direction of replication forks, while the blue signal represented the unmodified DNA (*Figure 3A*). Origins were positioned at the centre of two divergently oriented replication forks. DNA replication parameters, including replication fork speed, IODs and fork asymmetry, were determined as previously described (*Stanojcic et al., 2016*). Measurements from three biological replicates were pooled and column statistics are shown in *Supplementary file 2*. The median replication fork velocity was 1.73 and 2.44 kb/min for PCF and BSF, respectively, while the median IOD was 152 and 213 kb for PCF and BSF, respectively (*Supplementary file 2*). The median speed of replication forks was 1.4 times higher, and the median IOD was 1.4 times longer in BSF compared to PCF cells, with these differences between the two cell types being statistically significant (*Figure 3B and C*). Moreover, we compared the frequency of asymmetric replication

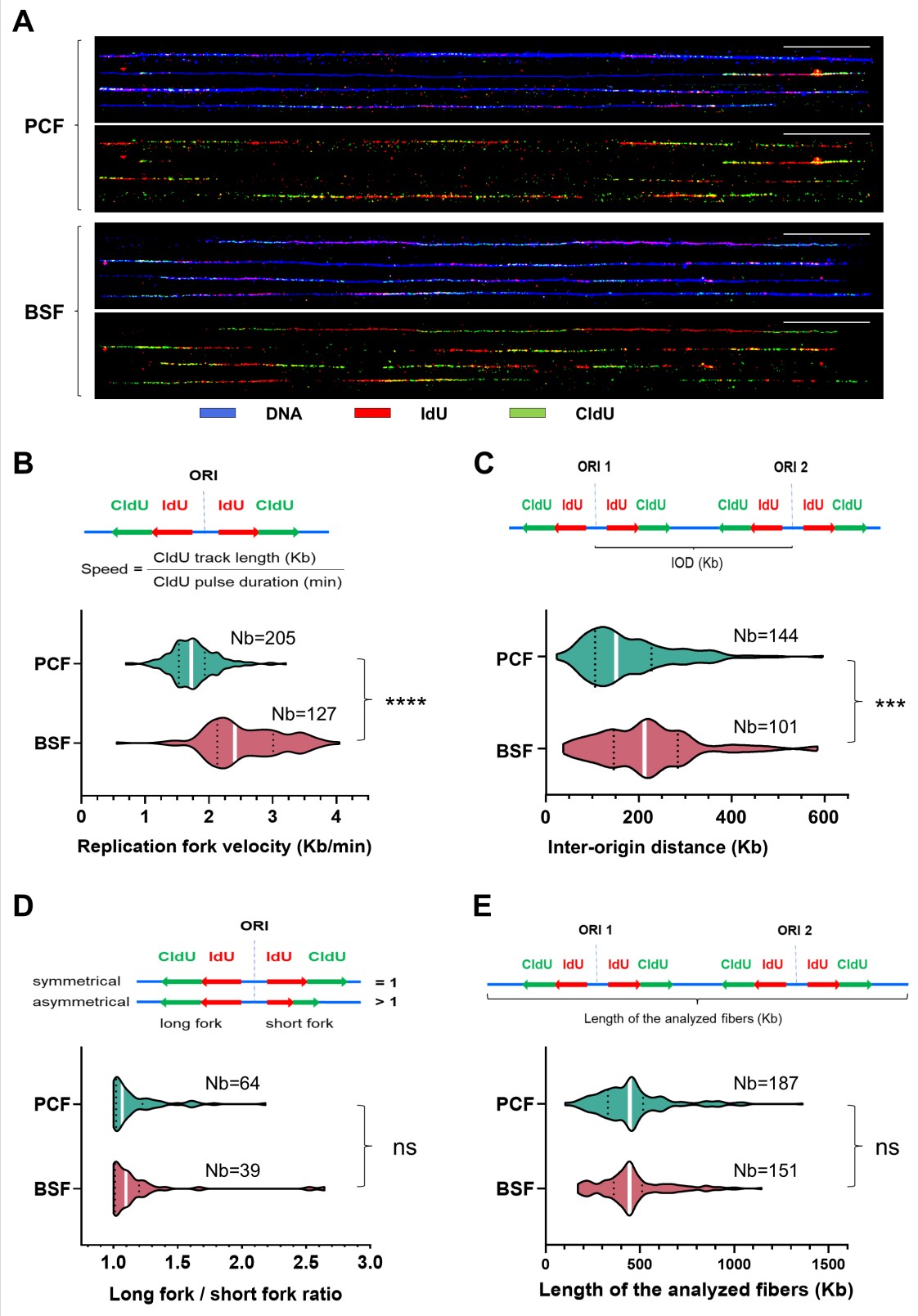

**Figure 3.** Comparison of DNA replication parameters between PCF and BSF cells by DNA molecular combing. (**A**) The figure depicts representative DNA molecules after immuno-detection. The immuno-detected DNA is indicated in blue, the initial pulse of nucleosides (IdU) in red, and the subsequent pulse of nucleosides (CldU) in green. The lower panels display only the red and green tracks from the first and second pulse of nucleosides extracted from the corresponding upper panels. 50 kb scale bars are shown as white lines. (**B**) The velocity of the replication forks was calculated by

*Figure 3 continued on next page*

*Figure 3 continued*

dividing the length of the CldU tracks with the duration of the CldU pulse on intact DNA molecules (schema on upper panel). Violin plots present distribution of the measured replication fork velocities in two cell types. (**C**) The inter-origin distance (IOD) was defined as the length between two adjacent replication initiation sites and can be determined by measuring the centre-to-centre distances between two adjacent progressing forks (schema on upper panel). Violin plots present the distribution of the measured IODs in two cell types. (**D**) The upper panel presents the concept of long fork/short fork ratios. The long fork/short fork ratio corresponds to the ratio of the longer IdU +CldU signal length over the shorter IdU +CldU signal length of bidirectional replication forks. A ratio >1 indicates fork asymmetry, while a ratio = 1 indicates fork symmetry (*Stanojcic et al., 2016*). The lower panel presents the distribution of the measured asymmetry of replication forks in two cell types. (**E**) The upper panel presents the concept of the measurements of the analysed DNA lengths. The lower panel presents violin plots with the measured lengths of the analysed DNA in two cell types. White bars on the violin plots indicate median values. Dotted black lines indicate quartiles. Two-tailed Mann-Whitney test was used to compute the corresponding p-values (ns - non-significant; * - p<0.05; ** - p<0.01; *** - p<0.001; **** - p<0.0001). The number of measurements (Nb) is given for each cell type.

forks, defined as bidirectional forks that progress from the origin at unequal rates, between the PCF and BSF cell types and found no statistically significant differences (*Figure 3D*). Given the high sensitivity of DNA combing measurements to the DNA fibre lengths (*Técher et al., 2013*), a comparison of the analysed DNA fibre lengths was performed as a control. This analysis showed that there was no statistically significant difference in the length of the analysed DNA molecules between the two cell types (*Figure 3E*, *Supplementary file 2*). This control confirmed that the observed differences in velocity and IODs were not an artefact of the different lengths of the analysed DNA molecules but rather reflected genuine differences between the replication of PCF and BSF cells.

The approximate number of active origins in a single cell can be estimated by dividing the length of genomic sequences (~50 Mb) by the mean or median IODs obtained by DNA combing. It was thus estimated that the range of active origins in a single cell was 279–329 for PCF and 228–235 for BSF cells (*Supplementary file 2*). The stranded SNS-seq method revealed 549 origins for PCF and 914 origins for BSF under stringent origin selection, which represent a fraction of all origins. Despite the inability of either method to determine the exact number of origins, the number of origins in a single cell tends to be smaller than in a population of cells. This phenomenon, known as flexible origin usage, is a well-documented feature of eukaryotic origins (reviewed in *Fragkos et al., 2015*).

## DNA replication starts between poly(dA) and poly(dT) enriched sequences

Having mapped a subset of origins of PCF and BSF cells, our next objective was to identify the common elements that would define the origins of *T. brucei*. To do so, we merged the origins of PCF and BSF cells into a single set, bringing the total number of origins to 1225. All analyses were also performed on the separate PCF and BSF sets of origins to demonstrate the similarity of their structures, and the results are presented in the figure supplements. As most origins were identified within intergenic regions, another shuffled control was developed to represent random intergenic regions selected in accordance with the number of origins, their size, and chromosomal location (Methods). The mean distance of origins and intergenic shuffled controls from the genes was found to be similar (745.2 bp for origins and 797.6 bp for shuffled controls), thus providing a more accurate control. Our first objective was to examine the nucleotide distribution of centred origins and compare it with the centred shuffled controls. The resulting plots showed that the percentage of adenines (A) or thymines (T) was higher near the centre of the origins (*Figure 4A*). This percentage was well above 40%, which was higher than the mean A or T content of the intergenic regions (reaching 30%; *Figure 4A*), while the overall A or T content of the genomic sequence is approximately 28%. The enrichment near origins showed a remarkable sequence polarity represented by a strong enrichment of A upstream and a strong enrichment of T downstream of the origin centre (*Figure 4A*, *Figure 4—figure supplement 1A*). Furthermore, a remarkable polarity in the distribution of guanines (G) and cytosines (C) was also observed. The origins exhibited a slight enrichment of G nucleotides upstream (up to 24%, while the average intergenic G was 20%), coinciding with the A enrichment, and a slight enrichment of the C nucleotides (up to 24%, while the average intergenic C was 20%), coinciding with the T enrichment downstream of the centres (*Figure 4A*). The nucleotide pattern that was specific for origins was entirely absent in the shuffled controls (*Figure 4A*, *Figure 4—figure supplement 1A*). The subsequent step was to search for consensus motifs of origins (Methods). However, apart from

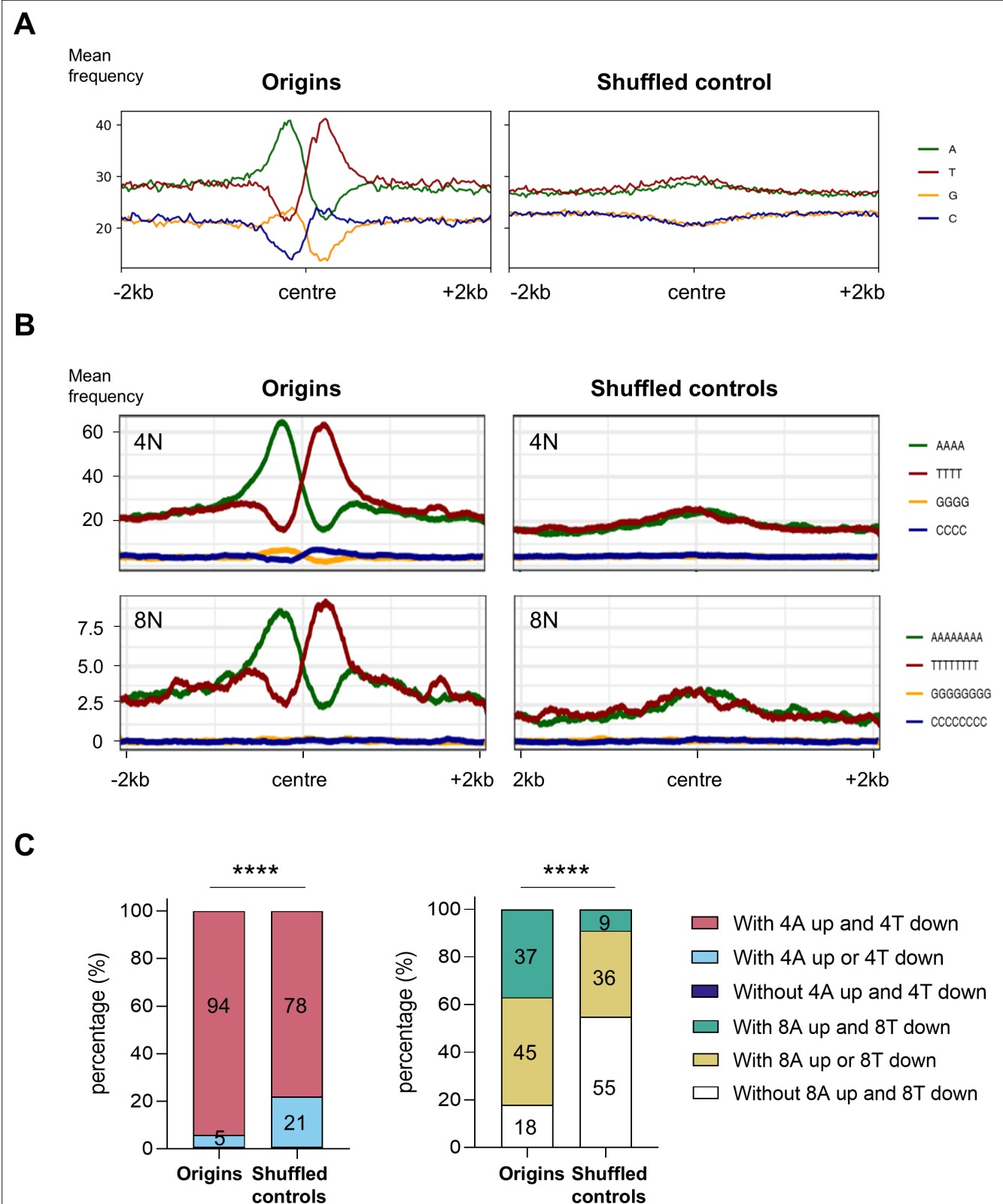

**Figure 4.** Spatial organisation of nucleotides and polynucleotides in the vicinity of origins. (**A**) The profile plots illustrate the distribution of four nucleotides around centred origins and shuffled controls (±2 kb). The nucleotide percentage was calculated within 20 bp window. (**B**) The profile plots illustrate the distribution of four and eight polynucleotides around centred origins and shuffled controls (±2 kb). A smoothing function was employed to calculate the mean frequency of polynucleotides per position within a 100 bp window (50 bp up- and downstream from the position). (**C**) The

*Figure 4 continued on next page*

*Figure 4 continued*

proportions of origins and shuffled controls without or with four or eight As upstream and/or four or eight Ts downstream of the centre. The ±0.5 kb window from the centre was analysed. The p-values were calculated using the Chi-square test, which involved a comparison of the absolute numbers of indicated categories for each pair of datasets (**** - p<0.0001).

The online version of this article includes the following figure supplement(s) for figure 4:

**Figure supplement 1.** Nucleotide and Polynucleotide distribution in the vincinity of origins.

**Figure supplement 2.** Nucleotide and Polynucleotide distribution in the vincinity of PCF and BSF origins.

**Figure supplement 3.** The profile plots show nucleotide distributions around two sets of regions derived from the same stranded SNS-seq experiments.

extended sequences that were enriched with adenines or thymines, no discernible consensus motif was identified (data not shown). The sequences surrounding the origins were also analysed using the WebLogo software (*Crooks et al., 2004*), and a graphical representation of nucleotide occurrence unambiguously confirmed an unequal nucleotide distribution, with an enrichment of A and G upstream and T and C downstream of the origins (*Figure 4—figure supplement 1B*). We then looked at the distribution of polynucleotides of different lengths around the origins. Poly(dA)-rich sequences were enriched upstream and poly(dT)-rich sequences downstream of the origins, a pattern absent in shuffled controls (*Figure 4B*, *Figure 4—figure supplement 1C*). Interestingly, the upstream region showed an enrichment of up to four consecutive Gs, while the downstream region showed an enrichment of up to four consecutive Cs (*Figure 4B*). We also checked nucleotide and polynucleotide distributions around PCF and BSF origins separately and found that the distributions were similar in both cell types and equivalent to the merged set of origins (*Figure 4—figure supplement 2A and B*). Accordingly, an origin can be defined as a genomic site with a specific uneven distribution of nucleotides, characterised by an overrepresentation of poly(dA)s and poly(dG)s upstream, and poly(dT)s and poly(dC)s downstream of the origin centre.

Subsequently, the proportions of origins that lacked or possessed four As upstream and/or four Ts downstream of the centre were calculated and compared with the intergenic shuffled controls (*Figure 4C*, left graph). The same calculation was performed for eight As and Ts (*Figure 4C*, right graph). The findings revealed that 1149 origins exhibited the presence of four As upstream and four Ts downstream of the centre, representing a proportion of ~94% compared to ~78% for the intergenic shuffled controls. Similarly, 450 origins (~37%) possessed eight As upstream and eight Ts downstream of the centre, compared to only 114 (~9%) intergenic shuffled controls with this pattern. Accordingly, the distribution of poly(dA) and poly(dT) observed for origins was found to be significantly different from that observed for the intergenic shuffled controls (*Figure 4C*).

## G4 structures accumulated in a specific pattern in proximity to the origin

G4s, DNA secondary structures with non-canonical Watson-Crick base pairing, have recently emerged as potential regulators of genome replication (*Valton and Prioleau, 2016*). Given the previously reported presence of G4 structures in the vicinity of some eukaryotic origins (*Besnard et al., 2012*; *Valton et al., 2014*; *Comoglio et al., 2015*; *Prorok et al., 2019*), we investigated the correlation between G4s and origins of *T. brucei*. To do this, we used two sets of G4s that had previously been determined by G-quadruplex sequencing (G4-seq; *Marsico et al., 2019*). One set of G4s was obtained under physiological conditions, while the other was obtained in the presence of pyridostatin (PDS), a drug that binds and stabilises G4s (*Marsico et al., 2019*). The SNS-seq origin dataset was intersected with stranded G4-seq datasets obtained in both conditions (Methods), resulting in a notable enrichment of G4s in the vicinity of the origins (*Figure 5A*, *Figure 5—figure supplement 1A*). A distinctive pattern of G4 distribution was observed, whereby the G4s on the plus strand were found to be enriched upstream, while the G4s on the minus strand exhibited an enrichment downstream of the centres of origin (*Figure 5A*, *Figure 5—figure supplement 1A*). This pattern of G4 enrichment was absent in the intergenic shuffled controls (*Figure 5A*, *Figure 5—figure supplement 1A*). The occurrence of G4s near the origins varied, with a discernible increase of G4s near the centre observed under PDS-stabilised conditions in comparison to physiological conditions (*Figure 5A*, *Figure 5—figure supplement 1A*).

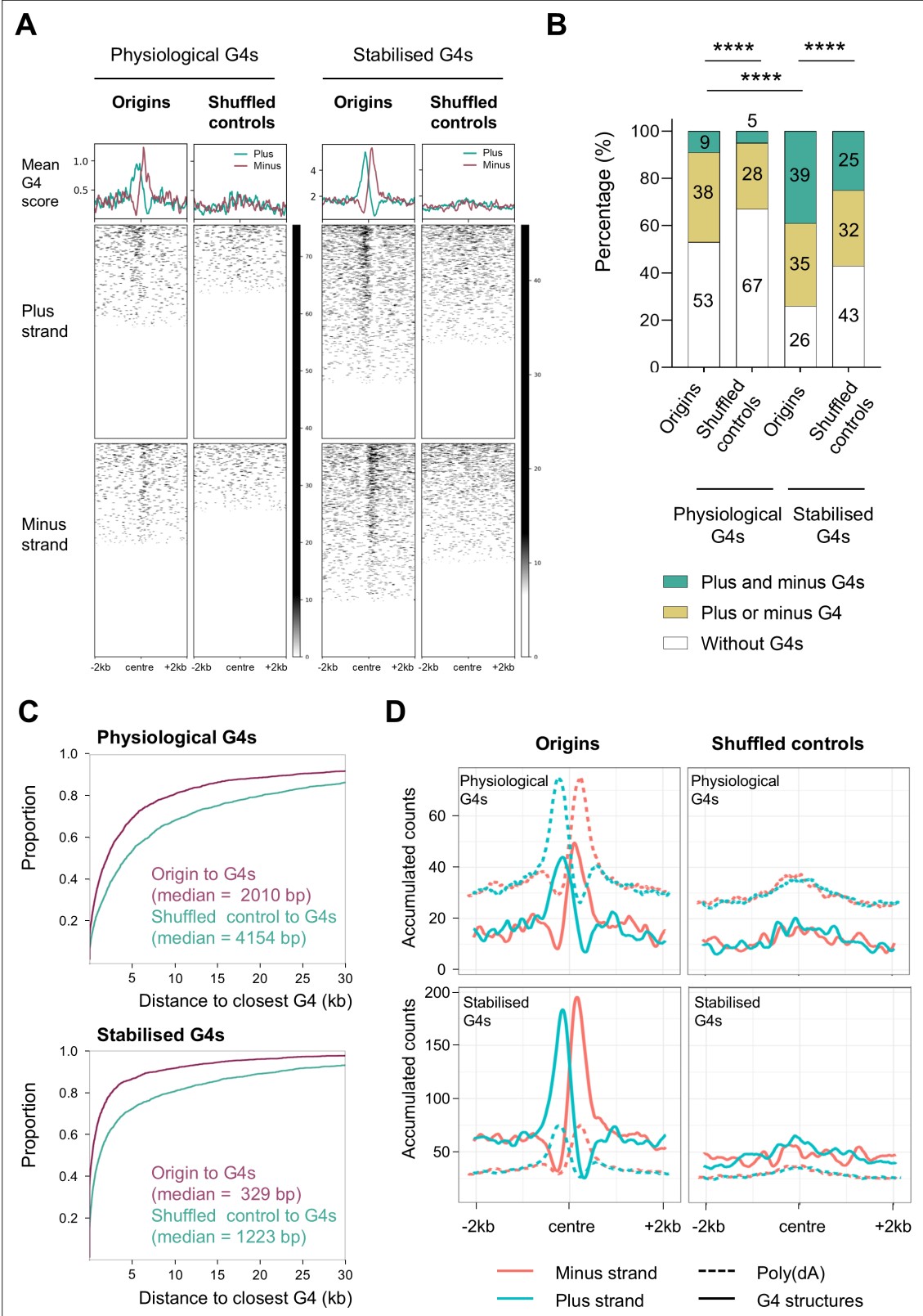

**Figure 5.** Distribution of G4 structures in the vicinity of origins. (**A**) The profile plots and heat maps show the distribution of the experimentally obtained G4 structures around centred origins and intergenic shuffled controls (±2 kb) in the Tbb TREU927 reference genome (Methods). The plus strand (light blue) and minus strand (pink) G4s, obtained under physiological conditions and in the PDS drug stabilised condition (***Marsico et al., 2019***) were overlapped with origins mapped by stranded SNS-seq. Mean G4 score presents average G4 score (***Marsico et al., 2019***) per 20 bp window.

*Figure 5 continued on next page*

*Figure 5 continued*

(**B**) The proportions of origins that lack or possess G4 structures on one or both sides of the centre were determined. The ±2 kb window from the centre was analysed for the presence of G4s. Two sets of experimental G4 structures (***Marsico et al., 2019***) were subjected to analysis in comparison with intergenic shuffled controls. The p-values were calculated using the Chi-square test, which involved a comparison of the absolute numbers of indicated categories for each pair of datasets (**** - p<0.0001). (**C**) Empirical Cumulative Distribution Function (ECDF) of the distances between the origins and the closest physiological and stabilised G4 structures (***Marsico et al., 2019***). Median distances are indicated. (**D**) The profile plots illustrate the distribution of the poly(dA) sequence (AAAA; dashed lines) and the experimental G4s (physiological and PDS drug stabilised; ***Marsico et al., 2019***) (solid lines) around centred origins and intergenic shuffled controls (±2 kb). The analysis was conducted in the Tbb TREU927 reference genome (Methods). The plus strand is represented in blue and the minus strand in pink. A smoothing function was employed to calculate the accumulated counts of the AAAA and G4s per position within a 100 bp window (50 bp up- and downstream from the position).

The online version of this article includes the following figure supplement(s) for figure 5:

**Figure supplement 1.** Distribution of G4 structures in the vincinity of PCF and BSF origins.

Our subsequent objective was to determine the proportion of origins that possessed G4s on one or both sides and those that lacked G4s. To achieve this, the region of ±2 kb around the centres of origins was analysed and compared to shuffled controls. The result showed that the origins displayed significantly different G4 distributions compared to the shuffled controls (p<0.0001; ***Figure 5B***). Furthermore, the proportions were significantly different in PDS-stabilised compared to physiological conditions (***Figure 5B***). A set of origins with G4s on both sides constituted a minor population under physiological conditions (9%), whereas this was a major population in PDS-stabilised conditions (39%). The proportion of origins with G4 on one or both sides increased significantly in PDS-stabilised condition (74%), compared to the physiological condition (47%; ***Figure 5B***). Analyses of the presence of G4s in the region ±0.5 kb around the centres of origins yielded comparable results (***Figure 5—figure supplement 1B***). Quantification of G4 distribution around the PCF and BSF origins was similar in the two cell types, and comparable to the merged set of origins (***Figure 5—figure supplement 1C***).

To gain a deeper insight into the distribution of origins and the G4 structures, an empirical cumulative distribution function (ECDF) was computed. The ECDF demonstrated that the distances between the origins and the closest G4s were smaller than those observed for the shuffled controls (for physiological G4s condition - median of 2010 bp for origins versus median of 4154 bp for intergenic shuffled controls and for stabilised G4s condition – median of 329 bp for origins versus median of 1223 bp for intergenic shuffled controls; ***Figure 5C***). Half of the origins had a stabilised G4 structure within ±329 bp of the centre, while 80% of the origins had the same structure within ±2575 bp of the centre. Therefore, a strong association was identified between origins and physiological G4 structures (odds ratio = 1.87; p-value = 0), as well as for stabilised G4s structures (odds ratio = 2.13; p-value=0).

Our results demonstrated that poly(dA) enriched sequences and G4s were enriched upstream of origins on the plus strand and downstream on the minus strand (***Figures 4B and 5A***). Therefore, we wanted to determine the spatial relationship between these two elements. To achieve this, the strand separated poly(dA) enriched sequences and the strand separated experimental G4s (***Marsico et al., 2019***) were intersected at centred origins and shuffled controls (Methods). The results demonstrated that the peaks of the experimental G4s and poly(dA) sequences overlapped near the centres of origins in a strand-specific manner that was notably absent in the intergenic shuffled controls (***Figure 5D***). Our findings revealed a previously unidentified element of origin: the enrichment of poly(dA) and G4 structures exhibited an important association upstream of the centre on the plus strand and downstream of origin on the minus strand (***Figure 5D***). It was observed that the poly(dA) peaks exhibited summits between ±100–300 nt, while the G4 peaks exhibited summits in the range of ±100–200 nt from the centres of origins (***Figure 5D***). Such regions were designated as poly(dA)/G4-enriched elements of origins.

## The origins exhibited varying nucleosome occupancy

Previous studies demonstrated the presence of origins in nucleosome-depleted regions (***Lipford and Bell, 2001***; ***Berbenetz et al., 2010***; ***Foss et al., 2024***; ***Poulet-Benedetti et al., 2023***) and their positioning adjacent to well-positioned nucleosomes on one or both sides (***Eaton et al., 2010***; ***Berbenetz et al., 2010***; ***Foss et al., 2024***; ***Poulet-Benedetti et al., 2023***; ***Jansen and Verstrepen, 2011***). Therefore, our subsequent objective was to determine whether there was a correlation between the origins identified by stranded SNS-seq and the previously published MNase-seq datasets, which described

the distribution of nucleosomes in *T. brucei* (*Maree et al., 2017*). After intersecting the origins and nucleosome positioning datasets (Methods), we identified a nucleosome signature specific to origins that was absent in the shuffled control. Origins showed a distinct 'M' pattern of nucleosome distribution around the centre, with varying nucleosome occupancy (*Figure 6A*, *Figure 6—figure supplement 1A*). The centre of origin displayed a markedly lower nucleosome occupancy (LNO) in comparison to the adjacent regions. Additionally, the centre of origin was situated between two regions exhibiting a higher nucleosome occupancy (HNO) in comparison to the shuffled controls (*Figure 6A*, *Figure 6—figure supplement 1A*).

We then quantified the number of origins without or with HNO regions on one or both sides of the centre, and the presence or absence of LNO in the centre and compared them to the intergenic shuffled controls (Methods). We performed this quantification for the merged set of origins (*Figure 6B*), but also for the mapped origins in PCF and BSF cells against four replicates of the PCF and four replicates of the BSF MNase-seq datasets (*Maree et al., 2017*; *Figure 6—figure supplement 1B*). Six categories of different combinations of HNO and LNO were analysed, and quantification showed that there were statistically significant differences between origins and corresponding shuffled controls when PCF and BSF origins were merged (*Figure 6B*) and when PCF and BSF origins were analysed separately (*Figure 6—figure supplement 1B*). We then analysed each of the six categories individually to determine which had changed significantly between the origins and the shuffled controls. It was found that three categories were significantly different between origins and shuffled controls, while three categories were not (*Figure 6C*). The category two HNO flanking LNO was significantly enriched (twice more), whereas the category no HNO with LNO and the category no HNO no LNO were significantly depleted in the origins compared to the shuffled controls (*Figure 6C*). Therefore, the active origins had significantly more LNO flanked by two HNO, at the expense of the no HNO with LNO and the no HNO no LNO categories.

Further investigation was undertaken to determine the spatial relationship between poly(dA)/G4 enriched sequences and nucleosome distribution. To achieve this, strand separated poly(dA) enriched sequences and predicted G4s were overlapped with the MNase-seq data (*Maree et al., 2017*; Methods). The plot obtained after the intersections revealed that the LNO region was positioned between the plus and minus poly(dA)/G4 summits (*Figure 6D*). Additionally, the HNO peaks were found to exhibit overlap with the poly(dA) and G4 peaks and to surround the origins (*Figure 6D*). Furthermore, these intersections permitted the estimation of the averaged distance of the above-mentioned elements from the centre of origins. The LNO region extended to approximately ±100–150 nt from the centre, thereby defining an LNO area of approximately 200–300 bp in the centre of origin. The two HNO regions were located between 150 and 500 nt upstream and downstream from the centre of origin (*Figure 6D*).

## The origins colocalised with a subset of DRIP-seq enrichment sites

DRIP-seq is a widely used technique for genome-wide mapping of RNA:DNA hybrids. This method is based on the immuno-precipitation of fragmented genomic DNA by the S9.6 monoclonal antibody, which specifically recognises RNA:DNA hybrids (*Boguslawski et al., 1986*; *Wahba et al., 2016*). A recent study employed DRIP-seq in *T. brucei* BSF cells, revealing enrichment within intergenic regions, between coding sequences of polycistronic transcription units, and coinciding with sites of polyadenylation and nucleosome depletion (*Briggs et al., 2018*). Considering the established role of short RNA primers in initiating DNA synthesis at the leading and lagging strands, an examination of the colocalization of the origins and previously published DRIP-seq data (*Briggs et al., 2018*) was conducted (Methods). A significant accumulation of RNA:DNA hybrids was identified around the origins, with two distinct summits located close to the origin centre (*Figure 7A*). An enrichment was also evident in the vicinity of the centre of the intergenic shuffled controls (*Figure 7A*), which is consistent with previous reports indicating a high prevalence of RNA:DNA hybrids (R-loops) in intergenic regions (*Briggs et al., 2018*). However, this peak was observed to be lower in comparison to the peak observed near centres of origins. Furthermore, the two upper summits were absent in the shuffled controls (*Figure 7A*). Next, we quantified the number of origins and shuffled controls that overlapped with R-loops (*Briggs et al., 2018*). The results demonstrated that 90% of the origins and 68% of the shuffled controls exhibited overlap with the DRIP-seq enrichment (*Figure 7B*); a statistically significant correlation was observed (odds ratio = 4.17; *p*-value $2.87E^{-142}$), suggesting a link between

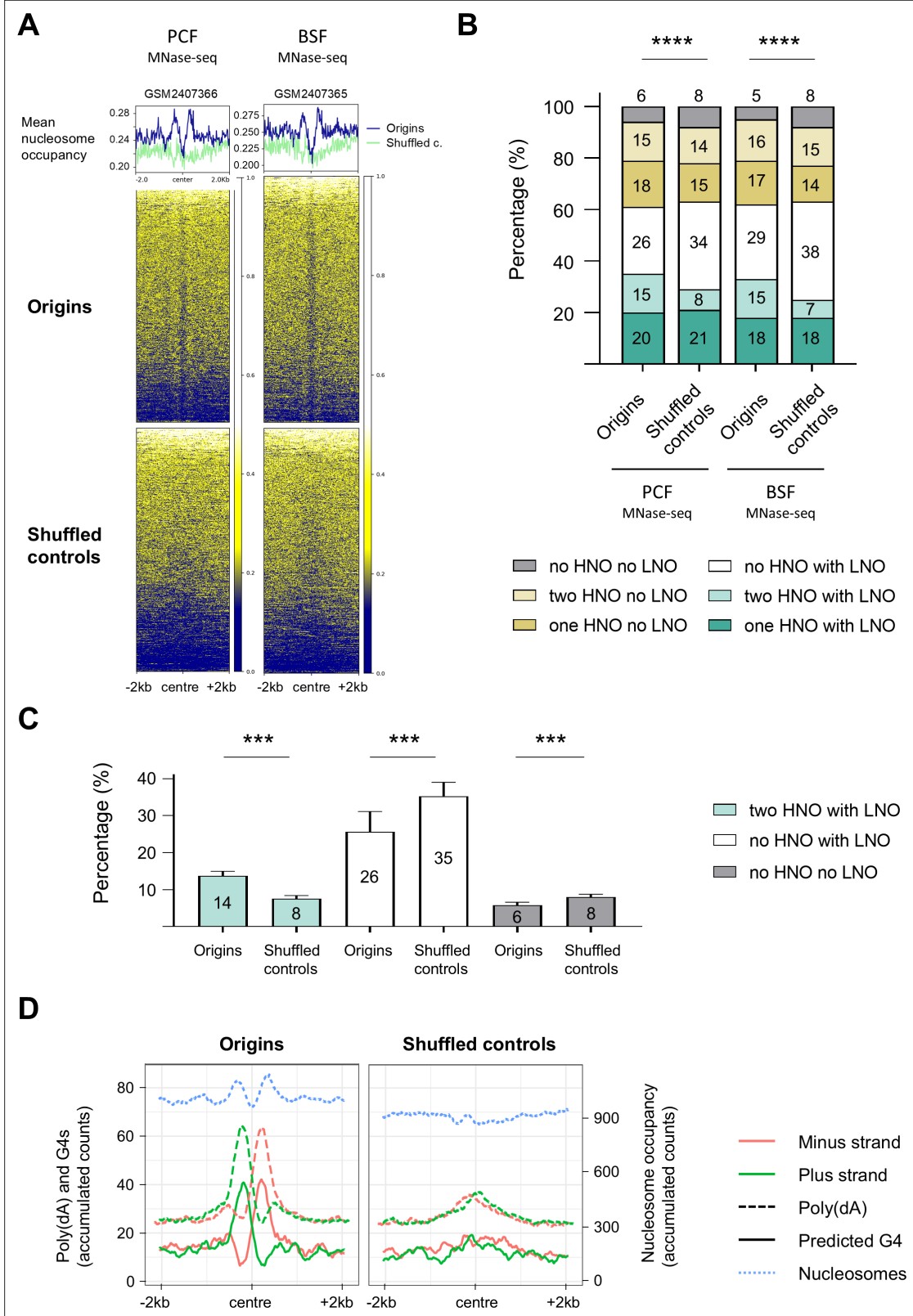

**Figure 6.** The distribution of nucleosomes around the origins in *T. brucei*. (**A**) The profile plot and heatmaps show the distribution of nucleosomes (MNase-seq data) detected in PCF and BSF cells (*Maree et al., 2017*) around the centred origins (blue line) and the intergenic shuffled controls (green line; ±2 kb) in Tb Lister 427 reference genome. One replicate of MNase-seq data (*Maree et al., 2017*) for PCF (GSM2407366) and one replicate for BSF cells (GSM2407365) is shown here, the other replicates are presented in *Figure 6—figure supplement 1A*. Mean nucleosome occupancy presents

*Figure 6 continued on next page*

*Figure 6 continued*

average nucleosome score (dyad value; *Maree et al., 2017*) per 20 bp window. (**B**) The percentage of origins and intergenic shuffled controls with or without high nucleosome occupancy (HNO) and low nucleosome occupancy (LNO) regions in the indicated combinations. Quantification was performed for the same replicates as indicated in the Figure 6A. The p-values were calculated using the Chi-square test, which involved a comparison of the absolute numbers of indicated categories for each pair of datasets (**** - p<0.0001). (**C**) Percentage of the three specified categories of HNO and LNO combinations that were significantly different between origins and shuffled controls. The remaining three categories were not statistically significant. The numbers indicate mean percentage values. The bars indicate standard deviations. Mann Whitney two-tailed test was performed to calculate p-values (*** - p<0.001). (**D**) The profile plots show the distribution of the poly(dA) sequence (AAAA; dashed line), the predicted G4 structures (solid line) and the nucleosome distribution of the replicate GSM2407365 from BSF cells (*Maree et al., 2017*; blue dotted line) around centred origins and intergenic shuffled controls (±2 kb). The G4s were predicted by the G4Hunter application (*Brázda et al., 2019*) in the Tb Lister 427 reference genome. The plus strand is represented in green, and the minus strand in pink. The values on the plots present the accumulated counts of the AAAA, G4s and nucleosome occupancy scores per position within a 100 bp window (50 bp up- and downstream from the position).

The online version of this article includes the following figure supplement(s) for figure 6:

**Figure supplement 1.** Distribution of nucleosomes around PCF and BSF origins.

RNA:DNA hybrid formation and origin activity. As DRIP-seq was only performed on BSF cells (*Briggs et al., 2018*), we intersected this set of data with PCF and BSF origins separately and found that the distribution of RNA:DNA hybrids around BSF origins was equivalent to the merged set of origins (*Figure 7—figure supplement 1A*). The distribution of RNA:DNA hybrids was also enriched around the centres of PCF origins with one of the upper summits less pronounced (*Figure 7—figure supplement 1A*). Quantification of SNS-seq and DRIP-seq overlap was performed and it was shown that 86% of PCF origins and 93% of BSF origins overlapped with R-loops (*Figure 7—figure supplement 1B*). It is worth noting that these intersections also revealed that only a small proportion of the previously identified RNA:DNA hybrids (R-loops; *Briggs et al., 2018*) overlapped with the merged origins (1.7%), PCF origins (0.7%), or BSF origins (1.3%).

Subsequently, the predicted G4 structures and poly(dA) enriched sequences from the plus and minus strands were superimposed with the DRIP-seq dataset (*Briggs et al., 2018*) at the centres of origins and shuffled controls (*Figure 7C*). This intersection allowed us to estimate the relative positions of these elements and their averaged distance from the centre of origins. The two distinct summits of R-loops were observed to be situated, on average, between the centre and 200 nt upstream and downstream from the centre of origins (*Figure 7C*).

## Comparison of the origins identified using stranded SNS-seq with previously published MFA-seq and TbORC1/CDC6 binding sites datasets

We compared the replication origins identified by stranded SNS-seq with the data obtained by Marker frequency analysis coupled with deep sequencing (MFA-seq; *Tiengwe et al., 2012*; *Devlin et al., 2016*). MFA-seq compares read coverage between replicating and non-replicating cells and identifies regions with increased DNA coverage in S phase, representing replicated regions. In *T. brucei*, MFA-seq identified broad regions of a few hundred kilobases (*Tiengwe et al., 2012*; *Devlin et al., 2016*), whereas the origins detected by stranded SNS-seq are about 150 bp long. We used six MFA-seq datasets (*Tiengwe et al., 2012*; *Devlin et al., 2016*; Methods) and we presented core chromosomal overview of the MFA-seq regions (*Tiengwe et al., 2012*; *Devlin et al., 2016*), TbORC1/CDC6 binding sites (*Tiengwe et al., 2012*) and the stranded SNS-seq origins on the 11 core chromosomes of *T. brucei* (*Figure 8A*). We found that 28–42% of the origins identified by stranded SNS-seq overlapped with early and 43–55% overlapped with late S-phase MFA-seq replicated regions (*Figure 8B*). We also reanalysed the overlap between TbORC1/CDC6 binding sites (*Tiengwe et al., 2012*) and MFA-seq regions (*Tiengwe et al., 2012*; *Devlin et al., 2016*). Our findings indicated that this overlap is in a similar range as the overlap between SNS-seq origins and MFA-seq replicated regions (*Figure 8B*).

Several proteins have been identified as potential subunits of the origin recognition complex (ORC) in *T. brucei*. One of these is TbORC1/CDC6, which complemented the yeast protein Cdc6, but not Orc1, in a phenotypic complementation assay (*Godoy et al., 2009*). We intersected the origins detected by stranded SNS-seq with the previously detected binding sites of the TbORC1/CDC6-12Myc tagged protein (*Tiengwe et al., 2012*; Methods). For this intersection, we used the data from TriTrypDB that contains 6482 ORC1/CDC6 binding sites, we filtered this dataset

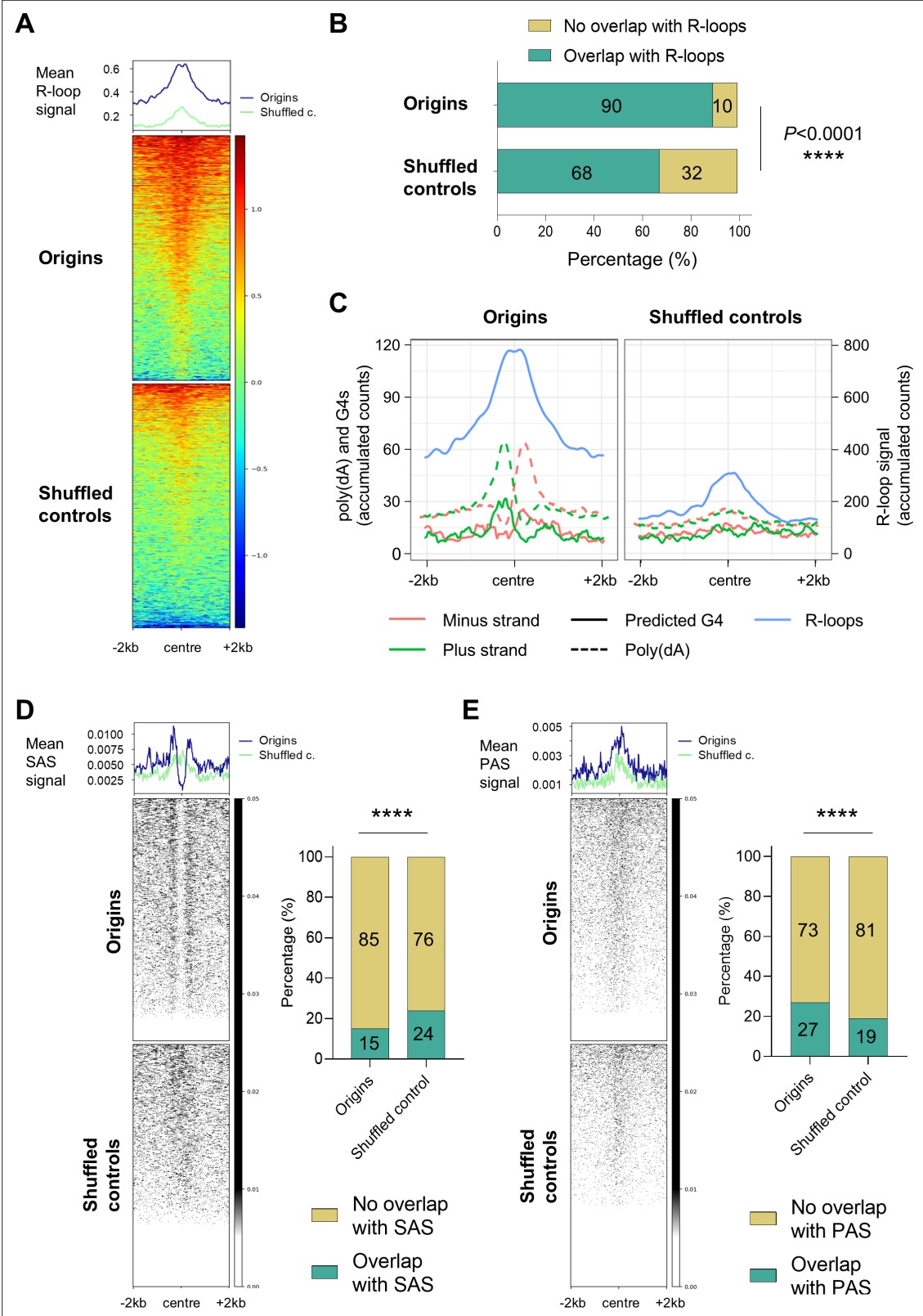

**Figure 7.** The distribution of DRIP-seq enrichment, splice acceptor sites (SAS), and polyadenylation sites (PAS) around the origins. (**A**) The profile plots and heatmaps show the distribution of R-loops (*Briggs et al., 2018*) around the centred origins (blue line) and the intergenic shuffled controls (green line; ±2 kb). Mean DRIP-seq signal presents average DRIP-seq signal (*Briggs et al., 2018*) per 20 bp window. (**B**) The proportion of origins and shuffled controls that overlap with R-loops (intersection without window). The p-values were calculated using Fisher's exact (two-sided) test, which involved

*Figure 7 continued on next page*

*Figure 7 continued*

a comparison of the numbers of indicated categories for each pair of datasets (\**** - p<0.0001). (**C**) The profile plots illustrate the distribution of the stranded poly(dA) sequence (AAAA; dashed line), predicted G4s (solid line) and R-loops (*Briggs et al., 2018*; solid blue line) around centred origins and shuffled controls (±2 kb). The plus strand is represented in green, and the minus strand in pink. The 13,409 G4s were predicted by the G4Hunter application (*Brázda et al., 2019*) in the Tb Lister 427–2018 reference genome. The values on the plots present the accumulated counts of the AAAA, G4s and R-loop signals per position within a 100 bp window (50 bp up- and downstream from the position). (**D**) Left panel: the profile plots and heatmaps show the distribution of transcription splice acceptor sites (SAS) around the centred origins (blue line) and the intergenic shuffled controls (green line; ±2 kb). Intersection was performed in the Tbb TREU927 reference genome for the 11 megabase chromosomes. Right panel: the proportion of origins and shuffled controls that overlap with SAS (intersection without window). The p-value was calculated using Fisher's exact (two-sided) test, which involved a comparison of the numbers of indicated categories for each pair of datasets (\**** - p<0.0001). (**E**) Left panel: the profile plots and heatmaps show the distribution of transcription polyadenylation sites (PAS) around the centred origins (blue line) and the intergenic shuffled controls (green line; ±2 kb). Intersection was performed as in Figure 7D. Right panel: the proportion of origins and shuffled controls that overlap with PAS. The p-values were calculated using Fisher's exact (two-sided) test, which involved a comparison of the numbers of indicated categories for each pair of datasets (\**** - p<0.0001).

The online version of this article includes the following figure supplement(s) for figure 7:

**Figure supplement 1.** Distribution of RNA:DNA hybrids around the centred origins of PCF and BSF cells.

at score 22 to obtain 990 binding sites which is close to the 953 TbORC1/CDC6 binding sites described previously (*Tiengwe et al., 2012*; Methods). The intersection revealed that TbORC1/CDC6 levels are lowest at the centre of origins (*Figure 8C*). We observed that 12.2% of SNS-seq origins overlap with 24.2% of TbORC1/CDC6 binding sites within a ±2 kb window, 17% of SNS-seq origins overlap with 24.2% of ORC1/CDC6 binding sites within a ±3 kb window and 26.9% of SNS-seq origins overlap with 53.4% of ORC1/CDC6 binding sites in ±5 kb window. Therefore, we identified a depletion of TbORC1/CDC6 binding sites at the centre of the origins, accompanied by a slight accumulation of ORC1/CDC6 binding sites 5 kb upstream and downstream of the origins (*Figure 8C*).

## Spatial coordination between the activity of the origins and transcription

DNA replication in trypanosomatids occurs in a particularly challenging environment, as most of their genomes are constitutively transcribed. Arrays of genes oriented in the same direction are transcribed by RNA polymerase II into polycistronic transcription units (PTUs), which can contain hundreds of genes (reviewed in *Clayton, 2019*). The primary transcript is processed into individual mRNAs by trans-splicing of a capped 'spliced leader' at the splice acceptor site (SAS) and by polyadenylation at the polyadenylation sites (PAS; reviewed in *Clayton, 2019*). To gain insight into the relationship between origins and SAS and PAS, we intersected our dataset with the unified SAS and PAS datasets for the core chromosomes (Methods). The intersection of origins with SAS showed that the centre of origins was depleted of SAS, while some origins had SAS flanking the centre and the SAS signal was enriched ±200–500 nt from the centre of origins (*Figure 7D*, left panel). Only 15% of the origins overlapped with SAS compared to 24% of the shuffled controls, and this difference was statistically significant (*Figure 7D*, right panel). The intersection of the origins with PAS revealed an accumulation of PAS near the origin centres. A similar, although less pronounced, accumulation was also observed in the shuffled controls (*Figure 7E*, left panel). Quantification showed that 27% of origins overlapped with PAS compared to 19% of shuffled controls and this difference was statistically significant (*Figure 7E*, right panel).

PTUs are separated by strand-switching regions (SSRs) and head-to-tail (HT) regions. SSRs are classified as either divergent (dSSRs) or convergent (cSSRs). dSSRs and cSSRs serve as transcription start (TSS) and termination sites (TTS), respectively (reviewed in *Clayton, 2019*). Our results showed that the origins are predominantly located in the intergenic regions (*Figure 2C and D*). The metaplots revealed that the TSS, TTS, and HT regions (*Kim, 2019*) were not enriched with origins identified by stranded SNS-seq (*Figure 8D*). Only 3.7% of origins are located within 30% of TSS, TTS, and HT regions. Therefore, most origins are localised in actively transcribed regions, which could lead to collisions between DNA replication and the transcription machinery. This spatial coincidence implies that transcription and replication must occur in a highly ordered and cooperative manner in *T. brucei*.

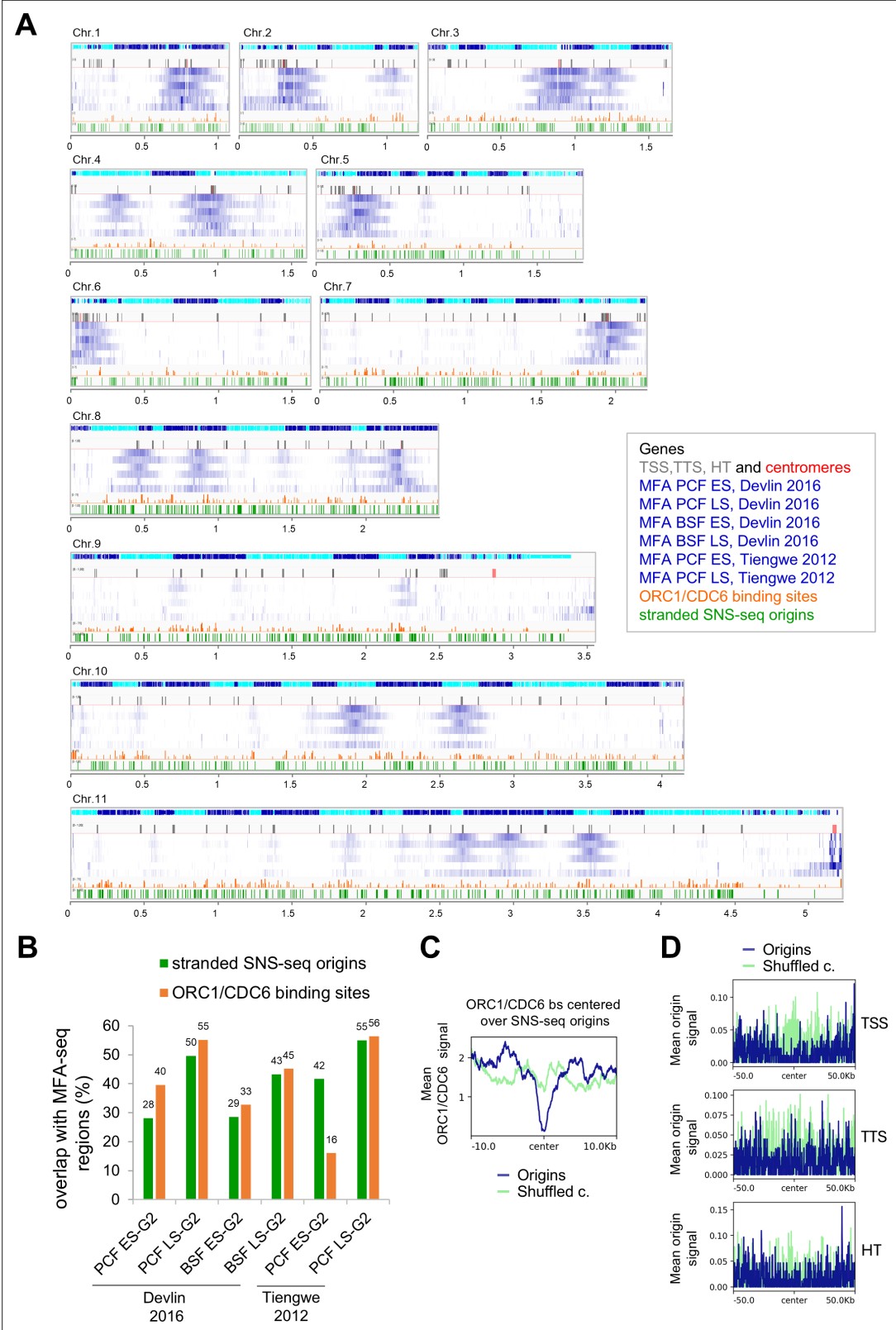

**Figure 8.** Comparison of stranded SNS-seq origins with MFA-seq replicated regions and TbORC1/CDC6 binding sites. (**A**) Chromosomal overview of MFA-seq and stranded SNS-seq origins from Integrative Genomics Viewer (IGV) (scale in Mb). The 11 core chromosomes of the Tb Lister 427–2018 reference genome are shown. The first track shows genes and their orientation (blue presents genes on the plus strand and turquoise presents genes on the minus strand). The second track shows centromeres in red (for chromosomes 9–11 the centromeres are not mapped to the core chromosome)

*Figure 8 continued on next page*

*Figure 8 continued*

and in grey TSS, TTS, and HT regions 56. Tracks 3–8 show MFA-seq data 42, 43 displaying the ratio of read depth between G2 and S phase in blue (scale 1–1.8). Tracks 3 and 4 show MFA-seq data from PCF cells (track 3 - early S phase to G2 phase ratio and track 4 - late S phase to G2 phase ratio). Tracks 5 and 6 show MFA-seq data from BSF cells (track 5 - early S phase to G2 phase ratio and track 6 - late S phase to G2 phase ratio) 43. Tracks 7 and 8 show MFA tracks from PCF cells. (Track 7 represents the early S phase to G2 phase ratio and track 8 represents the late S phase to G2 phase ratio) 42 (data obtained from Richard McCulloch via personal communication). Track 9 shows ORC1/CDC6 binding sites (retrieved from TriTrypDB) in orange. Track 10 shows the positions of stranded SNS-seq origins in green. (**B**) Bar plot presenting the percentage of the overlap of stranded SNS-seq origins (in green) and ORC1/CDC6 binding sites (in orange) with MFA-seq regions from the indicated data sets. Overlap was calculated without a window of 100 nt. The first 4 bars are overlap from *Devlin et al., 2016* 43 data; the last two bars are overlap from *Tiengwe et al., 2012* 42 data. Overlap of stranded SNS-seq origins with early S phase to G2 phase MFA data sets are shown in light blue and overlap with late S phase to G2 phase MFA data sets are shown in dark blue. (**C**) Profile plots illustrating the distribution of Orc1/Cdc6 binding sites 42 around centred origins (±2 kb upper panel and ±10 kb lower panel) for merged origins (PCF and BSF) in blue and corresponding shuffled control in light green. The mean Orc1/Cdc6 signal was calculated within 20 bp windows. (**D**) Profile plots illustrating the distribution of merged origins (PCF and BSF) in blue and corresponding shuffled control in light green around centred TSS (upper panel), TTS (middle panel), and HT regions (lower panel) 56 (±50 kb). The mean origin signal was calculated within 20 bp windows.

## Discussion

We used stranded SNS-seq to identify a set of active origins in the unicellular eukaryote, *T. brucei*. Our methodology is comparable to previously published origin-derived single-stranded DNA sequencing (Ori-SSDS; *Pratto et al., 2021*) but includes few differences. The stranded library preparation process was different in these two approaches and the SNS-depleted control was included in the origin mapping process as a background control. The key benefit of stranded over conventional (non-stranded) SNS-seq is the ability to identify origins with greater specificity by eliminating peaks that do not align with the proper orientation of SNS of an active origin. In addition, this method maps the origins between two divergent SNS peaks, allowing the determination of the centre of origins or replication initiation sites, which was essential for the discovery of the positions and orientations of the elements involved in the topology of origins. Origins mapped between two divergent SNS peaks showed the origin-specific nucleotide distribution around the centre (*Figure 4A*, *Figure 4—figure supplement 3A*). On the other hand, if we apply conventional SNS-seq origin mapping to our data and map origins in the middle of peaks without knowing their orientation, the origin-specific nucleotide distribution is lost (*Figure 4—figure supplement 3B*). Therefore, the primary benefits of our approach include the ability to centre origins between two divergent SNS peaks and to eliminate false positive peaks.

Using this stringent origin selection procedure, we have selected and analysed a subset of all origins. Some weak or inefficient origins may be missed due to low activity, resulting in low read coverage. It is therefore very likely that the number of origins of this parasite is higher than the number reported here. We also examined the origins separately in the two life stages demonstrating that the origin structures in the PCF and BSF cells were equivalent and justifying the merging of these two sets of origins into one. Under this stringent origin mapping condition, we found that 238 origins are in common (*Figure 2—figure supplement 2C*). However, we decided not to compare the PCF and BSF sets of origins in more detail because comparing non-exhaustively mapped origins in these two life cycle stages could be misleading rather than indicative of true differences between the two cell types.

Our analysis revealed no consensus sequence associated with the origins, which is consistent with the findings of most other studies of eukaryotic origins, including trypanosomatids (*Lombraña et al., 2016*; *de O Vitarelli et al., 2024*; *Tiengwe et al., 2012*; *Devlin et al., 2016*). Nevertheless, the genomic sites that function as origins are not random sequences. Thanks to the stranded SNS-seq approach, we were able to determine the positions and strand-specificity of some genetic elements and to present a comprehensive model of an origin (*Figure 9*). The results of our study showed that DNA replication starts at genomic sites where adenine and thymine are overrepresented and arranged in a specific pattern. On the plus strand, poly(dA) enriched sequences were situated upstream, while poly(dT) enriched sequences were located downstream of the origin's centre (*Figures 4B and 9*). The G4s near the centre of origins were embedded in poly(dA)-rich sequences and the summits of G4 peaks covered 100–200 nt up and downstream from the centres (*Figures 5D and 8*). Our model shows that SNS molecules are synthesised using poly(dA)/G4-rich complementary strands as templates (*Figure 9*). In our study, the enriched molecules detected in divergent peaks do not contain G4s. Rather, they are complementary to the strands enriched with G4s. Therefore, the

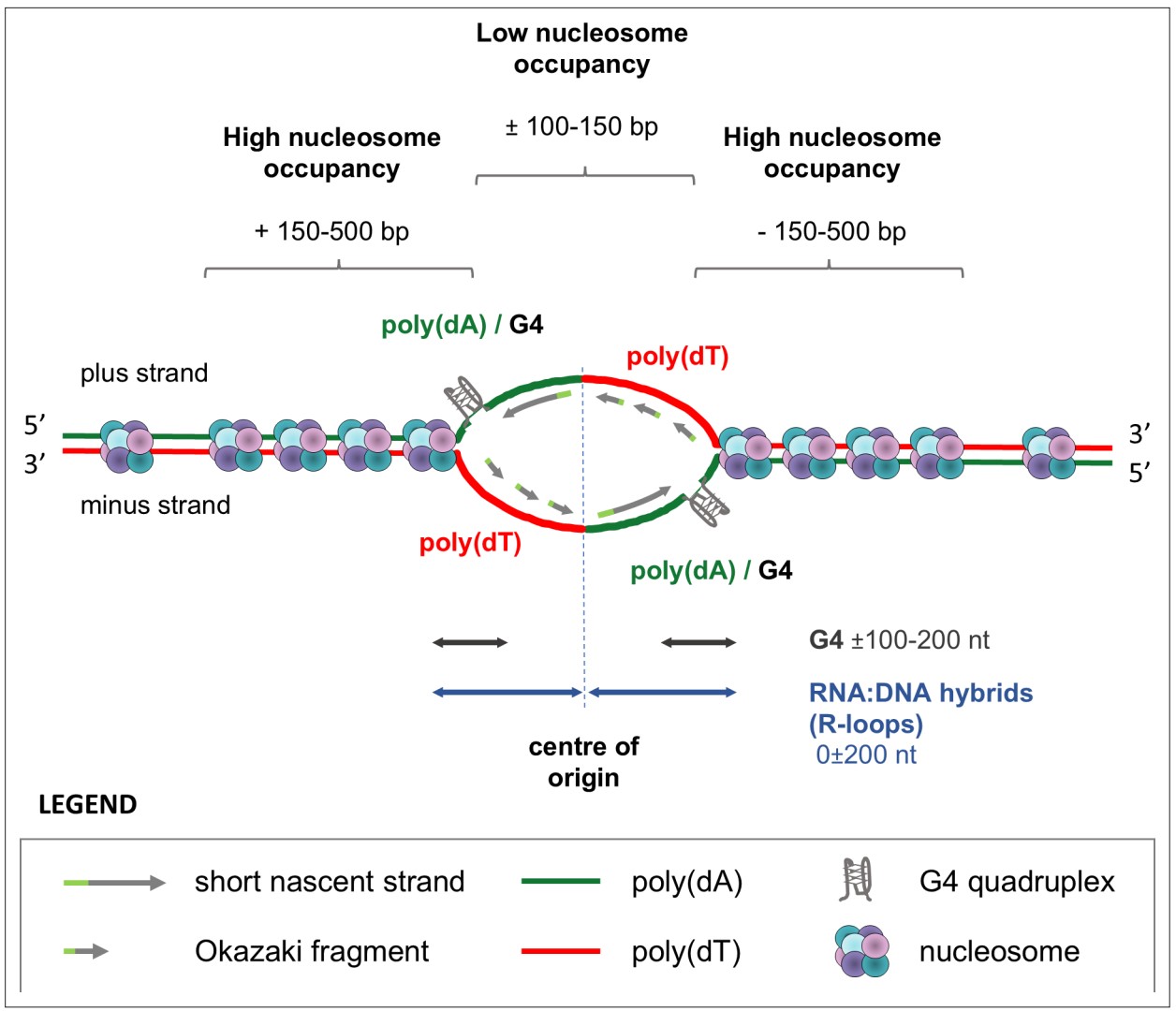

**Figure 9.** A proposed model of origin with the position of different genetic elements and nucleosome occupancy. Poly(dA) enriched sequences, interspersed with G4 (poly(dA)/G4) are enriched upstream of the origin centres on plus strand and downstream of origin centres on minus strand. Poly(dT) enriched sequences are enriched downstream of origin centres on plus strand and upstream on minus strand. The centre of the origin is a low nucleosome occupancy (LNO) region, flanked by high nucleosome occupancy (HNO) regions. The double arrow lines indicate the position of the summits of the peaks of different origin elements. The presented positions were calculated from the averaged distances. It should be noted that not all origins have the same spacing. The G4 structures, LNO and HNO regions were identified at a limited number of origins; however, they are illustrated on this model to demonstrate the potential of origins to form these structures.

observed enrichment of G4s around origins cannot be an artefact of the enzymatic treatments, as previously suggested (*Foulk et al., 2015*). The centres of origin were identified as LNO regions, which had an average length of 200–300 nt and were flanked by HNO regions located from ±150 to±500 bp from the centre (*Figures 6D and 9*). RNA:DNA hybrid enrichment revealed two distinct summits, one upstream and one downstream of the centre, with a range spanning ±200 nucleotides (*Figures 7A, C and 9*).

Our analysis indicated that G-rich sequences near origins tend to form G4 structures (*Figure 5A-C*, *Figure 5—figure supplement 1B-C*). The detection of more G4s under PDS-stabilised conditions compared to physiological ones suggests that there is an important potential for G4s formation near origins. G4 formation seems to be very transient and therefore not easy to detect unless stabilised by drug. It seems that G4 structures near origins undergo rapid changes, either forming or dissolving in response to the replication dynamics. We therefore propose that the formation of G4s plays a regulatory role in the activation of replication origins. Some of the recent publications suggested that

G4s may serve as replication fork barriers and play a role in transient replication fork stalling (*Comoglio et al., 2015*; *Zerbib and Simon, 2023*). The enrichment of G4 structures on strands acting as templates for leading strand synthesis implies that the formation of G4 structures could potentially pose or arrest replication fork progression.

Our analysis disclosed for the first time a strong link between RNA:DNA hybrid formation and DNA replication initiation. Given that RNA-primed nascent strands naturally form RNA:DNA hybrids during replication initiation, the enrichment of RNA:DNA hybrids near origins is both expected and biologically relevant. We found that a small proportion of previously reported RNA:DNA hybrids (1.7%) overlap with SNS-seq origins and we suggest that these RNA:DNA hybrids represent the SNS-derived RNA:DNA hybrids at the DNA replication initiation sites. The S9.6 antibody used for immunoprecipitation in DRIP-seq experiments exhibits a high affinity for RNA:DNA hybrids longer than 6 bp (*Bou-Nader et al., 2022*). The RNA:DNA hybrids are formed at the replication initiation sites by RNA priming of the SNS and Okazaki fragments and S9.6 antibody can immunoprecipitate these RNA:DNA hybrids. However, Okazaki fragments are not exclusive markers of origins as they form RNA:DNA hybrids throughout the genome, whereas RNA:DNA hybrids formed by RNA-primed SNS are specific markers of origins. Therefore, we suggest that the summits of RNA:DNA hybrids near the origins reflect the enrichment of RNA-primed SNS molecules. We also consider that the overlap between the two distinct techniques, DRIP-seq and stranded SNS-seq, provides compelling validation of our origin mapping approach.

The elements presented in our model were not found in all origins, as revealed by the quantification performed in this study. This may be caused by the fact that the datasets used for the intersections were obtained from four different strains, rather than from just one. Different strains are likely to activate different sets of origins, which could introduce bias during the intersection process. Another possible source of bias may be the varying quality of the genome assemblies, since this study used three different reference genomic sequences (Methods). This is clearly demonstrated by the detection of different numbers of origins in different reference genomic sequences using the same stranded SNS-seq data (Methods). An alternative hypothesis to explain the absence of certain structures in some origins is that they form transiently, only during origin activation. Therefore, it can be postulated that the presence of most of the elements, in addition to the chromatin state and nuclear organisation of origins, defines the moment of origin activation.

Different origin mapping methods often demonstrate limited overlap due to biases related to the chosen methodology (*Hyrien, 2015*; *Tian et al., 2024*). SNS-seq and MFA-seq are two complementary techniques that provide information on different aspects of DNA replication. SNS-seq is a widely used method of high-resolution origin mapping, whereas MFA-seq identifies replicated genomic regions in synchronised cells by comparing the number of DNA copies in the S phase and other phases of the cell cycle (*Tian et al., 2024*; *Marques et al., 2015*). The broad replicated regions (0.1–0.5 Mbp) identified by MFA-seq in *T. brucei* (*Tiengwe et al., 2012*) are likely to contain multiple origins, rather than just one. In this regard, we disagree with the interpretation of McCulloch's group, who postulated that there is a single origin per broad peak (*Tiengwe et al., 2012*). Our analysis shows that up to 28–55% of the origins detected by stranded SNS-seq locate within MFA-seq regions, providing complementary information on the position of origins within broad replicated regions. It should also be noted that the experimental context was different: MFA-seq relies on cell sorting, while stranded SNS-seq is performed on asynchronous cell populations, enabling the identification of origins active throughout S phase. A similar result was found when comparing conventional SNS-seq and MFA-seq in another trypanosomatid, *Leishmania major* (*Lombraña et al., 2016*).

We found a minority of TbORC1/CDC6 binding sites near origins detected by SNS-seq and this was comparable with the previously reported ~4.4% overlap of this protein with MFA-seq data (reviewed in *Marques and McCulloch, 2018*). Comparable results were also obtained in *T. cruzi* (*de O Vitarelli et al., 2024*). DNAscent nanopore sequencing revealed 4,287 origins in *T. cruzi*. Of these, 3782 (88.2%) did not match the 851 TcORC1/CDC6-Ty1 binding sites within a±3 kb window (*de O Vitarelli et al., 2024*). Therefore, only 11.7% of origins of *T. cruzi* overlapped with TcORC1/CDC6-Ty1 binding sites within a±3 kb window, a proportion similar to what we observed in *T. brucei* (6.2%). The origins detected by conventional SNS-seq in *Plasmodium falciparum* revealed 63% of overlap with initiation events detected by nanopore sequencing and 13.6% with PfORC$_{1-2}$ binding sites (*Castellano et al., 2024*). Another group found no significant correlation between origins detected by DNAscent

nanopore sequencing and PfORC1 binding sites within the same organism (*Totañes et al., 2023*). Furthermore, there is low overlap between origins mapped using different techniques and ORC-binding sites in human cells (*Hyrien, 2015*). We did not detect any accumulation of origins detected by stranded SNS-seq within SSRs. This differs from the previously reported accumulation of TbORC1/CDC6 binding sites at divergent SSRs in the Tb927 genome (*Tiengwe et al., 2012*). The reason for the low correlation between origins detected by different methods and ORC binding sites remains unknown, and further studies are necessary to answer this complex question.

It is important to note that the topology of origins of *T. brucei* is very similar to that previously described for other eukaryotes. The origins in mouse cells have been characterised as fragile sites, defined by long poly (dA:dT) tracks that are nucleosome free and devoid of the single-strand DNA-protecting protein RPA (*Tubbs et al., 2018*). The same position and orientation of predicted G4 motifs, peaking at 240–300 base pairs from replication start sites, was previously identified in two *drosophila* cell lines using conventional SNS-seq (*Comoglio et al., 2015*), as was found here for *T. brucei*. Furthermore, the organisation of nucleosomes around the origins of *T. brucei* is identical to that observed in yeast and higher eukaryotes (*Eaton et al., 2010*; *Lipford and Bell, 2001*; *Berbenetz et al., 2010*; *Foss et al., 2024*; *Poulet-Benedetti et al., 2023*; *Jansen and Verstrepen, 2011*; *Segal and Widom, 2009*; *Xu et al., 2012*). In the case of *L. major,* the origins identified using conventional SNS-seq were also located in genomic regions with lower nucleosome occupancy (*Lombraña et al., 2016*). In addition, a significant correlation between *L. major* origins and predicted G4 motifs was observed, with 74% of origins containing at least one such motif (*Lombraña et al., 2016*). These similarities provide strong support for the reliability of our origin identification in *T. brucei* and suggest that the structure of origins may be a conserved feature in all eukaryotes. However, further studies are needed to gain a deeper insight into the evolution of DNA replication origins.

## Methods

### Cell culture

Procyclic forms (PCF) of *T. brucei*, strain Lister 427 wild type strain were grown at 27 °C in SDM-79 (PAA Laboratories) supplemented with heat inactivated 10% FBS (Gibco) and 7 µg.mL$^{-1}$ hemin (*Wirtz et al., 1999*). Bloodstream forms (BSF) of *T. brucei* Lister 427 strain were cultivated at 37 °C, 5% CO in a modified Iscove's culture medium (Gibco) complemented according to *Hirumi and Hirumi, 1989*, with 10% (v/v) heat-inactivated foetal bovine serum (Gibco), 36 mM sodium bicarbonate (Sigma), 136 µg.mL$^{-1}$ hypoxanthine (Sigma), 39 µg.mL$^{-1}$ thymidine (Sigma), 110 µg.mL$^{-1}$ Na-pyruvate (Sigma), 28 µg.mL$^{-1}$ bathocuproine (Sigma), 0.25 mM β-mercaptoethanol, 2 mM L-cysteine.

### SNS isolation and purification

SNS were isolated and purified following a previously published protocol (*Cayrou et al., 2011*) with few modifications. Between 2 and 2.5x10$^9$ cells were used as starting material for SNS isolation. Total DNA was isolated from three biological replicates of asynchronous PCF and BSF cells in the exponential phase of growth (3–5x10$^6$ and 7–9 x10$^5$ cells.mL$^{-1}$ for PCF and BSF, respectively). Cells were precipitated and incubated in 0.5 mL of 20 mM Tris-Cl, pH 7.0, 50 mM EDTA, 1% N-Lauroylsarcosine, 0.5% Triton X-100 and 2 mg.mL$^{-1}$ proteinase K and incubated at 45 °C for 24 hr. Fresh solution of proteinase K was added after 24 hr and digested for 16 hr more. DNA was purified with phenol/chloroform/isoamyl alcohol and precipitated in 0.3 M Na-acetate (pH 5.3) and absolute ethanol. DNA was dissolved gently in TEN20 buffer (10 mM Tris-Cl, 50 mM EDTA, 20 mM NaCl, pH 7.0) with 100 U RNasin (Promega). DNA was heat-denatured (95 °C/5 min) and loaded onto 5 mL of neutral 5–20% sucrose gradients prepared in TEN20 buffer in ultra-clear tubes (Beckman Coulter). Size separation of nucleic acids was done by ultra-centrifugation in SW50 rotor for 7 hr at 45 krpm at +4 °C. Fractions of 200 µL from the sucrose gradient were collected and checked by agarose gel electrophoresis to assess the quality of nucleic acid separation (*Figure 10A*). Fractions from 0.5 to 2.5x10$^3$ nt were pooled together as SNS enriched fractions. Half of the SNS pool was treated with 10 µg of cocktail of RNase, DNase free (Roche, Cat Nb 11119915001) for 16 hr at 37 °C, to become the SNS-depleted control. SNS-enriched samples, together with its SNS-depleted controls, were treated three to four times with 50 U of T4 PNK (Thermo Fisher Scientific) and 100 U of $\lambda$ -exonuclease (Thermo Fisher Scientific). Each treatment was done after heat denaturation of DNA (95 °C/10 min) and simultaneously on samples and controls.

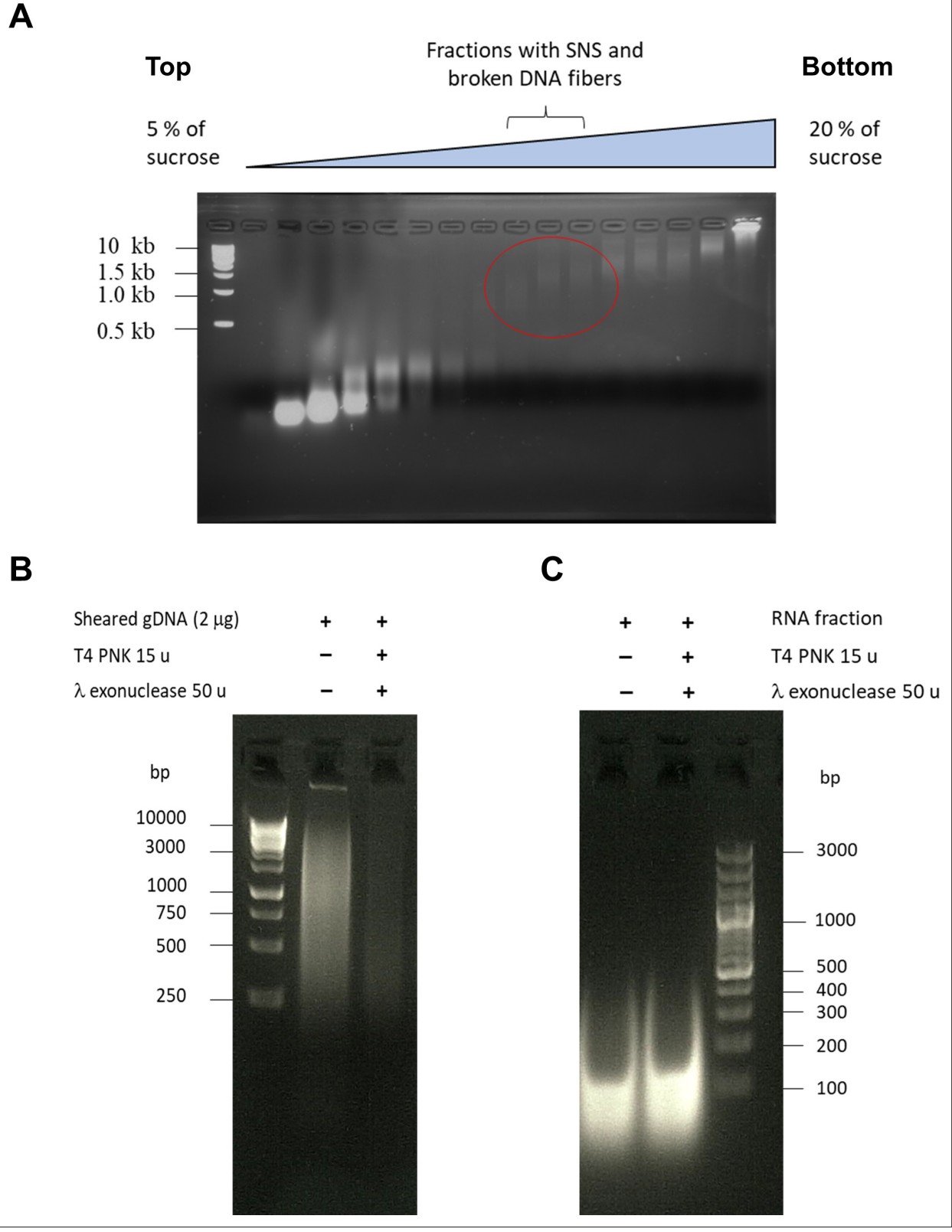

**Figure 10.** Isolation and validation of short nascent strands for stranded SNS-seq. (**A**) Native agarose gel electrophoresis of fractions from sucrose gradients. gDNA was denatured, centrifuged on 5–20% sucrose gradient and fractions were collected from the top (5% sucrose) to the bottom of gradient (20% sucrose). The size of DNA molecular marker is indicated on the right. Fractions containing nucleic acids of 0.5–2.5 kb are encircled. (**B**) Control of the efficiency of the T4 PNK and λ-exonuclease enzymes. gDNA was fragmented to the size of SNS fibres and 2 μg of starting gDNA

*Figure 10 continued*

was treated once with the indicated number of units of T4 PNK and $\lambda$-exonuclease enzymes to check the efficiency of enzymes. The SNS containing fractions were treated under the same condition but 3–4 times. The size of the DNA molecular marker is indicated. (**C**) RNA is preserved during treatment with T4 PNK and $\lambda$-exonuclease enzymes. RNA from the top fractions of the sucrose gradient was treated with the indicated number of units of T4 PNK and $\lambda$-exonuclease enzymes to check if the RNA fibres are preserved after this treatment. The size of the DNA molecular marker is indicated.

The online version of this article includes the following source data for figure 10:

**Source data 1.** Original files of the full raw uncropped, unedited gels.

**Source data 2.** Figures with the uncropped gels or blots with the relevant bands clearly labelled.

After denaturation, samples and controls were immediately placed on ice. The enzymatic reactions were assembled on ice, and then the enzymatic treatment with T4 PNK and $\lambda$-exonuclease was done at 37 °C. After enzymatic treatment and prior to library preparation, residual RNA was removed with 5 µg RNase, DNase-free (Roche Cat Nb 11119915001), isolated DNA was sonicated with Bioruptor Pico sonication device (Diagenode) and purified with magnetic beads (CleanNGS, Proteigen). The quantity of purified material was estimated by fluorometer DS 11 FX (DeNovix) using Qubit ssDNA Assay kit (Invitrogen).

It is important to note that we used SNS enrichment conditions and enzymatic treatments as described in previous reports (*Cayrou et al., 2011*; *Cayrou et al., 2012*). We enriched the SNS by size on a sucrose gradient and then treated this SNS-enriched fraction with high amounts of $\lambda$-exonuclease. A previous study has shown that under these conditions, complete digestion of DNA containing G4-rich sequences is achieved (*Cayrou et al., 2015*). The efficiency of the T4 PNK and $\lambda$-exonuclease enzymes was tested on both sheared gDNA and on RNA. The majority of DNA was digested after one treatment with these two enzymes (*Figure 10B*). RNA molecules treated under the same conditions were preserved (*Figure 10C*), suggesting that SNS would also be preserved. The quantity of DNA molecules in SNS containing fractions was systematically followed by qPCR after each enzymatic treatment and it was estimated that the amount of DNA in starting material was reduced around 100 times after the first treatment and around 10 times more after each subsequent treatment with these two enzymes.

## Preparation of stranded libraries for paired end sequencing

Single-stranded SNS libraries were prepared using the Accel-NGS 1 S Plus DNA Library Kit and 1 S Plus Dual Indexing Kit for Illumina (Swift Biosciences), following the manufacturer's protocol. A 3'end specific tail is preserving the strand orientation of the ssDNA fragments during library preparation, resulting in a second strand library, where the plus-strand fragments are represented by F1R2 read pairs and the minus-strand fragment by F2R1 read pairs. The quality of the libraries was checked by capillary electrophoresis on Bioanalyzer 2100 (Agilent) using Agilent High Sensitivity DNA Kit (Agilent Technologies). The quantity of libraries was estimated by fluorometry using dsDNA High Sensitivity Assay Kit on a DS-11 FX+ (DeNovix).

## Paired end sequencing (2x150)

The libraries were validated for quality and quantity using Fragment Analyzer and qPCR techniques. qPCR was performed using KAPA Library Quantification Kits - Complete kit (Universal; Roche, ref.7960140001) and Light Cycler 480 machine (Roche). The libraries were pooled equimolarly and sequenced on the Illumina NovaSeq 6000 using a NovaSeq Reagent Kit for 300 cycles (Illumina) following manufacturer's instructions. Libraries were sequenced on one lane of a SP flow-cell in paired-end 150 nt. Sequencing was performed by the GenomiX (MGX) Core Facility, Montpellier, France. Demultiplexing and FASTQ files production was done using the Illumina software bcl2fastq (RRID:SCR_015058, v2.20.0.422).

## SNS-seq data analysis

Quality control of the FASTQ files was performed using FastQC (RRID:SCR_014583, v0.11.9). The sequencing reads were tail trimmed to remove the 3'end specific tail introduced during library preparation using cutadapt *Martin, 2011* with the following parameters (-u –10 u 10 U –10 U 10). Afterwards the SNS-seq sequencing reads were aligned against the entire *T. brucei* Lister 427–2018 reference

genome release 55 from TriTrypDB (RRID:SCR_007043; , last accessed 24 June 2025) with bowtie2 (RRID:SCR_016368, v2.4.10; *Langmead et al., 2019*). The alignment output bam files were converted, sorted, and indexed using samtools (*Danecek et al., 2021*). Duplicated reads were removed using picard (RRID:SCR_006525, v2.23.5, 'Picard Toolkit', 2019. Broad Institute, GitHub Repository, last accessed 24 June 2025; Broad Institute). Mapped deduplicated reads pairs were then separated with samtools (RRID:SCR_002105, v1.13) *view* into minus-strand and plus-strand read pairs via their read orientation (minus-strand: F2R1 and plus-strand: F1R2). Peak calling was then performed with macs2 (v.2.2.7.1; using default parameters plus -p 5e-2 -s 130 m 10 30 `--gsize 2.5e7`) for plus and minus-strand reads separately with the SDC as control.

## Origin selection

Origins were selected after the peak calling step, during the peak filtering step. The called peaks were filtered using the bedtools suite (RRID:SCR_006646, v.2.30.0; *Quinlan and Hall, 2010*) as described below. In this peak filtering step, three criteria were defined to select the origins. The first criterion was that two SNS peaks must be oriented in a way that reflects the topology of the SNS strands within an active origin: the upstream peak must be on the minus strand and followed by the downstream peak on the plus strand. The second criterion was to set a maximal inner to inner border distance between two divergent minus and plus peak pair. The distances between the start points of the two divergent leading strands are not known. We conducted tests on several inner border-to-border distances, ranging from 0.5 to 2.5 kb. Higher numbers of origins were found with larger distances. However, we finally applied a maximum distance of 0.5 kb, presuming that this corresponds more closely to the physiological distance between two divergent SNS molecules. The third criterion was introduced to prevent complete overlap of two divergent peaks, as this can only occur if the two SNSs elongate at the same rate in both directions, which is an unlikely scenario. Indeed, the synthesis of the Okazaki fragment and ligation to the 5' of the SNS requires several enzymatic processes, and we can assume that the elongation of SNS fibre is slower at the 5' end. Of note, complete overlap of minus and plus peaks was rarely observed (mean 6.5% [stdev 6.5%, min 0.8%, max 15.8%]). To define the maximum overlap between the minus and plus SNS peaks, a series of tests were conducted, varying the percentage of overlap from 40% to 90%. This resulted in a similar number of origins, and ultimately, 50% was selected as the optimal value. Therefore, firstly, minus-strand and plus-strand peaks with an overlap of over 50% were removed with bedtools *intersect*. Secondly peak pairs on different strands with a maximal distance of 0.5 kb were selected using bedtools *windows*. Finally, peak pairs were filtered for the correct orientation (minus-strand followed by plus-strand) with bedtools *closest* and the origin was defined as the genomic region between the two inner borders of two divergent peaks. The overlap of origins between replicates and cell types as well as other genomic features was analysed with bedtools *windows* and bedtools *intersect*. Replicates were merged with bedtools *merge*.

For *visual inspection of genomic features*, integrative genomics viewer (IGV; RRID:SCR_011793; *Robinson et al., 2011*) was used. Violin plots were generated with the GraphPad Prism 8.3.0 software (RRID:SCR_002798, GraphPad Software Inc, La Jolla, CA, USA).

*Shuffled controls* were generated with bedtools *shuffle* (-chrom –noOverlapping –seed -excl) that randomly permutes genomic features provided respecting the number, the size, and chromosomal location and either excluding the detected origins or excluding detected origins and coding sequences (called intergenic shuffled control).

*Clustering of origins* origin clustering was analysed with bedtools *cluster* using different distances.

*The inter-origin distance (IOD)* between the origins detected by SNS-seq was calculated using bedtools *closest* (-io [ignore overlapping], -iu [ignore upstream]) on a single file.

Read sub-sampling was performed to be able to compare origin number in BSF and PCF. The mapped reads from all three PCF samples and control and BSF samples and control, respectively, were merged using Samtools merge. The PCF combined sample was sub-sampled with Samtools view (`--subsample` 0.69 `--subsample-seed` [0–14]) and the merged control with Samtools view (`--subsample` 0.57 `--subsample-seed` [0–14]) (*Cadoret et al., 2008*) times to obtain a set of mapped reads comparable to the BSF combined sample. Origin detection was applied as described above to the combined BSF and the [15] subsamples combined PCF samples.

For *visualization of the respective* localization of two or more datasets the Deeptools suite (RRID:SCR_016366, v3.5.1; *Ramírez et al., 2016*) and the ggplot2 R package was used. The open source ggplot2 R package (RRID:SCR_014601), freely available from the Comprehensive R Archive Network (CRAN, RRID:SCR_003005) repository (http://www.r-project.org, last accessed 24 June 2025) was used to generate composite plots. In Deeptools, bed and bigwig files were used to generate a matrix file with *ComputeMatrix* and *PlotHeatmap* or *PlotProfil* were used to generate visualizations from the matrix.

*Motive enrichments* were tested over the origin centres extended 250 bp up and downstream with the MEME suite motive discovery (RRID:SCR_001783; *Bailey and Elkan, 1994*) as well as with homer (RRID:SCR_010881) *findMotifs* (*Heinz et al., 2010*). A multiple sequence alignment and the analysis of nucleotide patterns were built with the WebLogo (RRID:SCR_010236) online application (*Crooks et al., 2004*).

*Nucleotide composition* around the origin centres was calculated as follows. The *T brucei* genome was split into 20 bp windows using bedtools *window*. The number of A, T, G, and C nucleotides for each window were obtained with bedtools *nuc* and used to calculate the nucleotide percentage per 20 bp window. The resulting bedgraph files were transformed into bigwig file format using USCS utilities *bedGraphToBigWi*g (https://github.com/ucscGenomeBrowser/kent/tree/master/src/utils, *ucscGenomeBrowser, 2026*, last accessed 24 June 2025).

## Comparison of origins mapped by stranded SNS-seq with previously published datasets

### Handling of datasets mapped to different reference genomes

The stranded SNS-seq data, that we mapped to the Tb Lister 427–2018 genome, were compared with various published datasets [SAS and PAS (TriTrypDB), MNase-seq (*Maree et al., 2017*), DRIP-seq (*Briggs et al., 2018*), G4-seq (*Marsico et al., 2019*), MFA-seq (*Tiengwe et al., 2012*), and Orc1/CDC6 ChIP-on-Chip (*Tiengwe et al., 2012*)] that were analysed using different references genomes (Tb Lister 427 or TREU927). To enable direct comparison, we either reanalysed the published datasets against the Tb Lister 427–2018 genome or remapped the stranded SNS-seq data to match the respective reference genomes of the compared datasets. The mapping of origins in the different reference genomes produced different numbers of origins, in Tb Lister 427 1059 PCF and 1466 BSF origins and in total 2081 origins were detected, in TREU927 976 PCF and 1466 BSF origins were detected and in total 1963 origins were detected. Specific details on the chosen approach are given in the relevant paragraphs below.

### Splice leader acceptor sites (SAS) and polyadenylation sites (PAS)

Splice leader acceptor sites (SAS) and polyadenylation sites (PAS) were obtained from TriTrypDB. The tracks 'unified spliced leader addition sites' and the 'unified poly A sites' were downloaded from the genome browser view for the 11 Megabase chromosomes of the Tbb TREU927 reference genome in bed file format. We compared these datasets to SNS-seq dataset analysed as described above in the Tbb TREU927 reference genome.

### Transcription start sites (TSS), transcription termination sites (TTS), and HT regions

A list of TSS, TTS, and head-to-tail (HT) regions for the Tbb TREU927 reference genome was obtained from *Kim, 2019*.

### MNase-seq

The MNase-seq data (*Maree et al., 2017*) was downloaded as bigwig files from GEO (accession number GSE90593). The data was published using the Tb Lister 427 reference genome. To be able to compare our SNS-seq dataset with this data, we reanalysed the SNS-seq sequencing reads as described above in the Tb Lister 427 reference genome (last accessed 24 June 2025). To quantify the MNase-seq signature we observed around the origin centres ('M' pattern), we extracted the average MNase-seq signal (normalised dyad values [*Maree et al., 2017*]) from the regions 400–200 nt upstream, 200–400 nt downstream and 75nt up- and downstream of the origin centre with the UCSC

utilities *bigWigAverageOverBed* command (here, **ucscGenomeBrowser, 2026**, last accessed 24 June 2025). Regions with negative values representing tandem repeats and N-rich sequences (**Maree et al., 2017**) were discarded. LNO regions were considered as present when the average MNase-seq value was below 0.23 normalised dyad value and the HNO regions were considered present when the averaged normalised dyad value for the region was above this value. This value corresponds to the average value for background regions (2000–1000 bp upstream of the origin centres and 1000–2000 bp downstream of origin centres).

## Drip-seq

The Drip-seq dataset was downloaded from ENA (accession number PRJEB21868; **Briggs et al., 2018**) as fastq files. The re-analysis was done as described (**Briggs et al., 2018**). The published DRIP-seq analysis was performed in Tb Lister 427 reference genome. We re-analysed the data in the Tb Lister 427 and the Tb Lister 427–2018 reference genomes. The intersection of DRIP-seq data with the stranded SNS-seq origins was done in the Tb Lister 427–2018 reference genome.

## G4-seq

The G4-seq (**Marsico et al., 2019**) dataset was downloaded from GEO (accession number GSE110582) as bed and bedgraph files separated for the plus and minus strand. G4-seq was performed on *T. brucei* EATR01125 strain and analysed in the Tbb TREU927 reference genome (**Marsico et al., 2019**) to be able to directly compare the data with the SNS-seq dataset we analysed the SNS-deq data in the Tbb TREU927 reference genome.

*G4 prediction* in the *T. brucei* genome were done with G4Hunter (**Brázda et al., 2019**) with the following settings, window size (-w 25) and a threshold of 1.57. Prediction of G4 structures in the Tb 427–2018 reference genome and in the Tb Lister 427 reference genome were performed, to be able to compare G4 structures to MNase-seq (in Tb Lister 427 reference genome) and DRIP-seq (in Tb Lister 427–2018 reference genome) data as G4-seq output files were available only for the Tbb TREU927 reference genome (**Marsico et al., 2019**).

## MFA-seq

We intersected six MFA-seq datasets. Two were obtained directly from the authors (**Tiengwe et al., 2012**) and four were downloaded from ENA (accession number PRJEB11437; **Devlin et al., 2016**), all in fastq file format. The analysis of the fastq files was performed as described previously (**Tiengwe et al., 2012**; **Devlin et al., 2016**). The original published MFA-seq analysis was performed in *T. brucei* Tbb TREU927 reference genome. We re-analysed the data in the Tbb TREU927 reference genome and in the Tb Lister 427–2018 reference genomes to be able to directly compare them to the stranded SNS-seq data.

## ORC1/CDC6 ChIP-on-chip

The Orc1/CDC6 ChIP-on-Chip dataset (**Tiengwe et al., 2012**) was downloaded from TriTrypDB (as ratios_peaks) for each element in the *T. brucei* Tbb TREU927 reference genome in form of bed files. The dataset available on TriTrypDB differs from the one published in the manuscript of **Tiengwe et al., 2012** but the originally published dataset is not available anymore (personal communication Richard McCulloch). The TriTrypDB ORC1/CDC6 dataset contains 6543 binding sites, whereas the dataset published originally contained 953 binding sites (**Tiengwe et al., 2012**). To obtain a number comparable to the published number of ORC1/CDC6 binding sites we filtered the TritrypBD data set at a score of 22 which resulted in a set of 990 ORC1/CDC6 binding sites. To be able to directly compare them to the SNS-seq, we analysed the SNS-seq data in the Tbb TREU927 reference genome.

## DNA molecular combing

Asynchronous cell populations of *T. brucei* were grown to the density of $4\times10^6$ and $7\times10^5$ cells.mL$^{-1}$ for PCF and BSF, respectively. Cells were sequentially labelled with two modified nucleosides, firstly with 300 µM iodo-deoxyuridine (IdU, Sigma) for 20 min and then 20 min with 300 µM chloro-deoxyuridine (CldU, Sigma) without intermediate washing. After labelling, the cells were immediately placed on ice

to stop DNA replication and extensively washed with ice-cold 1xPBS. Around 1x10$^8$ cells were blocked in 1% low melting agarose plugs.

DNA molecular combing was done as described previously (*Stanojcic et al., 2016*). Protein-free plugs with DNA were extensively washed in TEN20 buffer (10 mM Tris-Cl, 50 mM EDTA, 20 mM NaCl, pH 7.0) and then stained with 0.5 µM YOYO-1 fluorescent dye (Molecular Probes) for 1 hr at 37 °C. After washing from the excess YOYO-1 dye, the 200 µL of TEN20 buffer was added to the plugs, melted at 65 °C for 15 min, slowly cooled down to 42 °C and treated with 10 U of β-agarase (Merck) for 16 hr. This treatment was repeated with fresh β-agarase for 4 hr. After digestion, 2 mL of 50 mM MES (mix of 0.5 M MES Hydrate and 0.5 M MES Sodium Salt, pH 5.7) was added very gently to the DNA solution and then DNA molecules were combed and regularly stretched (2 kb/µm) on silanised coverslips provided by Montpellier DNA Combing Facility. Linearised DNA molecules were fixed for 16 hr at 65 °C, denatured in 1 N NaOH for 20 min and blocked with 1×PBS / 1% BSA / 0.1% Triton X-100. Immuno-detection was done with antibodies diluted in 1×PBS / 1% BSA / 0.1% Triton X-100 and incubated at 37 °C in a humid chamber for 60 min. Each step of incubation with antibodies was followed by extensive washes with 1×PBS. Immuno-detection was done with anti-ssDNA antibody (1/100 dilution, clone 16–19, Cat. No. MAB3034, Merck Milipore), the mouse anti-BrdU antibody (1/20 dilution, clone B44, Cat. No. 347580, Becton Dickinson) and rat anti-BrdU antibody (1/20 dilution, clone BU1/75, Cat. No. ABC117-7513, Abcys). The following secondary antibodies were used: Alexa488 Goat anti Rat IgG (1/50 dilution, Cat. No. A-11006, Invitrogen), Alexa 546 Goat anti Mouse IgG1 (1/50 dilution, Cat. No. A-21123, Invitrogen) and Alexa 647 goat anti-mouse IgG2a (1/100 dilution, Cat. No. 21241, Invitrogen). Coverslips were mounted with 15 µL of Prolong Gold Antifade (Cat. No. P10144, Thermo Fisher Scientific) and dried at room temperature for 16 hr. Images were acquired using Zeiss Z1 microscope, equipped with an ORCA-Flash 4.0 digital CMOS camera (Hamamatsu), using a 40×objective, where 1 pixel corresponds to 325 bp (one microscope field of view corresponds to ~450 kb). Acquisition and analysis of images were done by MetaMorph version 7 software (Molecular Devices LLC) (RRID:SCR_002368).

## Measurements and statistical analysis of combed DNA molecules

The speed of replication forks was estimated on individual forks, presented as IdU track followed by a CldU track. Only intact (non-fragmented) forks were measured, which was followed by DNA immuno-staining. The speed was calculated as the ratio between the CldU track length (kilobases) and the CldU pulse duration (minutes). The IODs were measured as the distance (in kilobases) between the centres of two adjacent progressing forks located on an uninterrupted DNA fibre. Fork asymmetry was calculated as the ratio of the longer track over the shorter track in progressing divergent forks. A ratio greater than one indicates asymmetry. The GraphPad Prism 8.3.0 software (GraphPad Software Inc, La Jolla, CA, USA) was used to generate graphs and to perform statistical analysis of DNA combing measurements as described previously (*Stanojcic et al., 2016*). The statistical comparisons of different datasets were assessed by the non-parametric Mann–Whitney two-tailed tests.

## Acknowledgements

Special thanks to Cedric Notredame for advising us to prepare SNS stranded libraries. We would like to thank Julie AJ Clement and Stephanie Le Gras for their help with preliminary analysis of SNS-seq data. We acknowledge the Montpellier DNA Combing Facility for providing silanised coverslips. We are grateful to MGX-Montpellier GenomiX (Montpellier, France) facility for performing sequencing. We would also like to acknowledge the 'Eukaryotic Pathogen, Vector and Host Informatics Resource' (VEuPathDB) for the integrated database, covering numerous eukaryotic pathogens (*Amos et al., 2022*). We are grateful to the Roscoff Bioinformatics platform ABiMS (http://abims.sb-roscoff.fr), part of the Institut Français de Bioinformatique (ANR-11-INBS-0013) and BioGenouest network, for providing computing resources. Thanks to Frédéric Bringaud (University Victor Segalen Bordeaux 2) for providing the *T. brucei* 427 wild-type PCF strain and Lucy Glover (Institute Pasteur, Paris) for providing us *T. brucei* 427 wildtype BSF strain. The work was supported by the Agence Nationale de la Recherche (ANR) within the frame of the "Investissements d'avenir" program [ANR-11-LABX-0024–01 "PARAFRAP" to YS], the Centre National de la Recherche Scientifique (CNRS), the French Ministry of Research and the Centre Hospitalier Universitaire of Montpellier. SS was recipient of a grant from the Fondation pour la Recherche Médicale

(ING20160435527). MGX acknowledges financial support from France Génomique National infrastructure, funded as part of "Investissement d'Avenir" program managed by Agence Nationale pour la Recherche (contract ANR-10-INBS-09). ABiMS is founded as a part of the Institut Français de Bioinformatique (ANR-11-INBS-0013).

## Additional information

### Funding

| Funder | Grant reference number | Author |
|---|---|---|
| Agence Nationale de la Recherche | ANR-11-LABX-0024-01 | Bridlin Barckmann Yvon Sterkers |
| FRM-Fondation pour la recherche Médicale | ING20160435527 | Slavica Stanojcic |
| Agence Nationale de la Recherche | ANR-10-INBS-09 | Simon George |
| Institut Français de Bioinformatique | ANR-11-INBS-0013 | Bridlin Barckmann |
| Agence Nationale de la Recherche | paraFrap | Bridlin Barckmann Yvon Sterkers |

The funders had no role in study design, data collection and interpretation, or the decision to submit the work for publication.

### Author contributions

Slavica Stanojcic, Investigation, Methodology, Writing – original draft, Project administration, Writing – review and editing; Bridlin Barckmann, Data curation, Software, Formal analysis, Validation, Methodology, Writing – original draft, Writing – review and editing; Pieter Monsieurs, Software, Methodology, Writing – original draft; Lucien Crobu, Simon George, Investigation; Yvon Sterkers, Conceptualization, Resources, Supervision, Funding acquisition, Validation, Methodology, Writing – original draft, Project administration, Writing – review and editing

### Author ORCIDs

Bridlin Barckmann ⓘ https://orcid.org/0000-0001-8354-668X
Yvon Sterkers ⓘ https://orcid.org/0000-0002-5623-5664

Reviewer #1 (Public review): https://doi.org/10.7554/eLife.108143.3.sa1
Reviewer #2 (Public review): https://doi.org/10.7554/eLife.108143.3.sa2
Author response https://doi.org/10.7554/eLife.108143.3.sa3

## Additional files

### Supplementary files

Supplementary file 1. Numbers of reads. The numbers of reads obtained after paired-end sequencing, numbers of reads, peaks and mapped origins after different steps of bioinformatic analysis are presented for three biological replicates of PCF and BSF cells. Each replicate contained an SNS-enriched sample and SNS-depleted control. (+) and (-) peaks called present the number of peaks called and localised on the plus or minus DNA strand, respectively. Filtered peaks (+) and (-) represent pairs of peaks with divergent orientation (first (-) followed by (+) peak) (Methods).

Supplementary file 2. DNA replication parameters obtained by DNA combing. DNA replication parameters obtained by DNA combing in two cell types (PCF and BSF) of *T. brucei*. The estimated numbers of origins were obtained by dividing the genomic sequence length (50.081 Mb) by either the median or the mean IODs calculated on combed fibres.

MDAR checklist

## Data availability

The SNS-seq data (FASTQ files) and the BED files containing the positions of the origins of the merged and separated PCF and BSF samples from this study have been submitted to the European Nucleotide Archive under the accession number PRJEB70708. The SNS-seq data analysis pipeline is available at https://github.com/bridlin/finding_ORIs (copy archived at *Barckmann, 2025*) and https://doi.org/10.5281/zenodo.14289282.

The following datasets were generated:

| Author(s) | Year | Dataset title | Dataset URL | Database and Identifier |
|---|---|---|---|---|
| Sterkers Y | 2024 | High-resolution mapping of the DNA replication initiation sites in *Trypanosoma brucei* | https://www.ebi.ac.uk/ena/browser/view/PRJEB70708 | European Nucleotide Archive, PRJEB70708 |
| Barckmann B | 2024 | bridlin/finding_ORIs: v1.0 | https://doi.org/10.5281/zenodo.14289282 | Zenodo, 10.5281/zenodo.14289282 |

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
