## [Editor Report · eLife Assessment]

The authors adapt sequencing of nascent DNA (DNA linked to an RNA primer, "SNS-Seq") to map DNA replication origins in *Trypanosoma brucei*. The main impact of this work is reporting a new set of putative origins, which do not overlap with previously reported origins, but which appear to overlap with previously mapped DNA-RNA hybrid (R-loops). Thus, these **valuable** findings open up new avenues for further investigation into the mechanistic basis for firing of replication forks in this organism. However, the supporting evidence remains **incomplete** and would benefit from orthogonal validation. This work will be of interest to those studying DNA replication and epigenetic regulation of fork origins.

---

## [Referee Report · Reviewer #1 (Public review)]

In this paper, Stanojcic and colleagues attempt to map sites of DNA replication initiation in the genome of the African trypanosome, *Trypanosoma brucei*. Their approach to this mapping is to isolate 'short-nascent strands' (SNSs), a strategy adopted previously in other eukaryotes (including in the related parasite Leishmania major), which involves isolation of DNA molecules whose termini contain replication-priming RNA. By mapping the isolated and sequenced SNSs to the genome (SNS-seq), the authors suggest that they have identified origins, which they localise to intergenic (strictly, inter-CDS) regions within polycistronic transcription units and suggest display very extensive overlap with previously mapped R-loops in the same loci. Finally, having defined locations of SNS-seq mapping, they suggest they have identified G4 and nucleosome features of origins, again using previously generated data. Though there is merit in applying a new approach to understand DNA replication initiation in *T. brucei*, where previous work has used MFA-seq and ChIP of a subunit of the Origin Replication Complex (ORC), there are two significant deficiencies in the study that must be addressed to ensure rigour and accuracy.

(i) The suggestion that the SNS-seq data is mapping DNA replication origins that are present in inter-CDS regions of the polycistronic transcription units of *T. brucei* is novel and does not agree with existing data on the localisation of ORC1/CDC6, and it is very unclear if it agrees with previous mapping of DNA replication by MFA-seq due to the way the authors have presented this correlation. For these reasons, the findings essentially rely on a single experimental approach, which must be further tested to ensure SNS-seq is truly detecting origins. Indeed, in this regard, the very extensive overlap of SNS-seq signal with RNA-DNA hybrids should be tested further to rule out the possibility that the approach is mapping these structures and not origins.

(ii) The authors' presentation of their SNS-seq data is too limited and therefore potentially provides a misleading view of DNA replication in the genome of *T. brucei*. The work is presented through a narrow focus on SNS-seq signal in the inter-CDS regions within polycistronic transcription units, which constitute only part of the genome, ignoring both the transcription start and stop sites at the ends of the units and the large subtelomeres, which are mainly transcriptionally silent. The authors must present a fuller and more balanced view of SNS-seq mapping, across the whole genome, to ensure full understanding and clarity.

In the revised manuscript, the authors have improved the presentation and analysis of their data, expanding the description of SNS-seq mapping across the genome, and more clearly assessing to what extent there is correlation between SNS-seq signal and previous mapping approaches to predict origins (by MFA-seq and ChiP-chip of ORC1/CDC6). With regard the correlation between SNS-seq and ORC/1CDC6 ChIP-chip, it should be noted that two datasets were generated in distinct strains of *T. brucei* (Lister 427 and TREU927, respectively), and it is unclear if the latter dataset can be accurately mapped to the strain used here. Notwithstanding this concern, these improvements clarify a number of aspects of the SNS-seq mapping: (1) the signal is more prevalent in the transcribed core of the genome than in the largely transcriptionally silent subtelomeres; and (2) whereas previous work revealed strong correlation between ORC1/CDC6 localisation and MFA-seq peaks at the ends of multigene transcription units, neither of these data show significant overlap with SNS-seq signal, which is not seen at transcription start or stop sites ('SSRs'; supplementary Fig.8D) and shows marked depletion at predicted ORC1/CDC6 sites (supplementary Fig.8C). To the authors' credit, they acknowledge this lack of correlation in the discussion.

The authors have not provided any new data to substantiate their assertion that SNS-seq accurately detects origins in *T. brucei*, and therefore the work rests on a single experimental approach, without validation. As a result, the suggestion of abundant, previously undetected origins in the intergenic regions of multigene transcription remains a prediction. One key untested limitation of the work lies in the observation that the very large majority of SNS-seq signal overlaps with previously RNA-DNA hybrids; without an experimental test, the suggestion that the authors have 'disclosed for the first time a strong link between RNA:DNA hybrid formation and DNA replication initiation' remains conjecture.

---

## [Referee Report · Reviewer #2 (Public review)]

Summary:

Stanojcic et al. investigate the origins of DNA replication in the unicellular parasite *Trypanosoma brucei*. They perform two experiments, stranded SNS-seq and DNA molecular combing. Further, they integrate various publicly available datasets, such as G4-seq and DRIP-seq, into their extensive analysis. Using this data, they elucidate the structure of origins of replications. In particular, they find various properties located at or around origins, such as polynucleotide stretches, G-quadruplex structures, regions of low and high nucleosome occupancy, R-loops, and that origins are mostly present in intergenic regions. Combining their population-level SNS-seq and their single-molecule DNA molecular combing data, they elucidate the total number of origins as well as the number of origins active in a single cell.

Between the initial submission and this revision, the raised major concerns have not been resolved, and no additional validation has been provided.

Strengths:

(1) A very strong part of this manuscript is that the authors integrate several other datasets and investigate a large number of properties around origins of replication. Data analysis clearly shows the enrichment of various properties at the origins, and the manuscript is concluded with a very well-presented model that clearly explains the authors' understanding and interpretation of the data.

(2) The DNA combing experiment is an excellent orthogonal approach to the SNS-seq data. The authors used the different properties of the two experiments (one giving location information, one giving single-molecule information) well to extract information and contrast the experiments.

(3) The discussion is exemplary, as the authors openly discuss the strengths and weaknesses of the approaches used. Further, the discussion serves its purpose of putting the results in both an evolutionary and a trypanosome-focused context.

Weaknesses:

I have major concerns about the origin of replication sites determined from the SNS-seq data. As a caveat, I want to state that, before reading this manuscript, SNS-seq was unknown to me; hence, some of my concerns might be misplaced.

(1) There are substantial discrepancies between the origins identified here and those reported in previous studies. Given that the other studies precede this manuscript, it is the authors' duty to investigate these differences. A conclusion should be reached on why the results are different, e.g., by orthogonally validating origins absent in the previous studies.

(2) I am concerned that up to 96% percent of all SNS-seq peaks are filtered away. If there is so much noise in the data, how can one be sure that the peaks that remain are real? Upon request, the authors have performed a control, where randomly placed peaks were run through the same filtering process. Only approximately twice as many experimental peaks passed filtering compared to random peaks. While the authors emphasize reproducibility between replicates, technical artifacts from the protocol would also be reproducible. Moreover, in other SNS-seq studies, for example, Pratto et al. Cell 2021, Fig. 1B, + and − strand peaks always appear closely paired. This pattern contrasts strongly with Fig. 2A in this manuscript.

Further, I have some minor concerns that do not affect the main conclusions of the manuscript:

- Fig 2C: The regions shown in the heatmap have different sizes, and I presume that the regions are ordered by size on the y-axis? If so, does the cone-shaped pattern, which is origin-less for genic regions and origin-enriched for intergenic regions, arise from the size of the regions? (I.e., for each genic region, the region itself is origin-less and the flanking intergenic regions contain origins.) If this is the case, then the peaks/valleys, centered exactly on the center of the regions on the mean frequency plots, arise from the different sizes of the analyzed regions, not from the fact that origins are mostly found at the center of intergenic regions. This data would be better presented with all regions stretched to the same size. This has not been addressed in the revision.

- Line 123, "and the average length of origins was found to be approximately 150 bp.": To determine origins, the authors filter away overlapping peaks and peaks that are too far from each other. Both restrict the minimal and maximal length of origins that can be observed, and this, in turn, affects the average length. This has not been addressed in the revision.

Are claims well substantiated?:

The identification of origins via SNS-seq appears to be incompletely supported to me.

All downstream analyses depend on the reliability of origin identification.

Impact:

This study has the potential to be valuable for two fields: In research focused on *T. brucei* as a disease agent, where essential processes that function differently than in mammals are excellent drug targets. Further, this study would impact basic research analyzing DNA replication over the evolutionary tree, where *T. brucei* can be used as an early-divergent eucaryotic model organism.

---

## [Author Response]

The following is the authors’ response to the current reviews.

**Public Reviews:**

**Reviewer #1 (Public review):**
In this paper, Stanojcic and colleagues attempt to map sites of DNA replication initiation in the genome of the African trypanosome, *Trypanosoma brucei*. Their approach to this mapping is to isolate 'short-nascent strands' (SNSs), a strategy adopted previously in other eukaryotes (including in the related parasite Leishmania major), which involves isolation of DNA molecules whose termini contain replication-priming RNA. By mapping the isolated and sequenced SNSs to the genome (SNS-seq), the authors suggest that they have identified origins, which they localise to intergenic (strictly, inter-CDS) regions within polycistronic transcription units and suggest display very extensive overlap with previously mapped R-loops in the same loci. Finally, having defined locations of SNS-seq mapping, they suggest they have identified G4 and nucleosome features of origins, again using previously generated data. Though there is merit in applying a new approach to understand DNA replication initiation in *T. brucei*, where previous work has used MFA-seq and ChIP of a subunit of the Origin Replication Complex (ORC), there are two significant deficiencies in the study that must be addressed to ensure rigour and accuracy.(i) The suggestion that the SNS-seq data is mapping DNA replication origins that are present in inter-CDS regions of the polycistronic transcription units of *T. brucei* is novel and does not agree with existing data on the localisation of ORC1/CDC6, and it is very unclear if it agrees with previous mapping of DNA replication by MFA-seq due to the way the authors have presented this correlation. For these reasons, the findings essentially rely on a single experimental approach, which must be further tested to ensure SNS-seq is truly detecting origins. Indeed, in this regard, the very extensive overlap of SNS-seq signal with RNA-DNA hybrids should be tested further to rule out the possibility that the approach is mapping these structures and not origins.(ii) The authors' presentation of their SNS-seq data is too limited and therefore potentially provides a misleading view of DNA replication in the genome of *T. brucei*. The work is presented through a narrow focus on SNS-seq signal in the inter-CDS regions within polycistronic transcription units, which constitute only part of the genome, ignoring both the transcription start and stop sites at the ends of the units and the large subtelomeres, which are mainly transcriptionally silent. The authors must present a fuller and more balanced view of SNS-seq mapping, across the whole genome, to ensure full understanding and clarity.In the revised manuscript, the authors have improved the presentation and analysis of their data, expanding the description of SNS-seq mapping across the genome, and more clearly assessing to what extent there is correlation between SNS-seq signal and previous mapping approaches to predict origins (by MFA-seq and ChiP-chip of ORC1/CDC6). With regard the correlation between SNS-seq and ORC/1CDC6 ChIP-chip, it should be noted that two datasets were generated in distinct strains of *T. brucei* (Lister 427 and TREU927, respectively), and it is unclear if the latter dataset can be accurately mapped to the strain used here. Notwithstanding this concern, these improvements clarify a number of aspects of the SNS-seq mapping: (1) the signal is more prevalent in the transcribed core of the genome than in the largely transcriptionally silent subtelomeres; and (2) whereas previous work revealed strong correlation between ORC1/CDC6 localisation and MFA-seq peaks at the ends of multigene transcription units, neither of these data show significant overlap with SNS-seq signal, which is not seen at transcription start or stop sites ('SSRs'; supplementary Fig.8D) and shows marked depletion at predicted ORC1/CDC6 sites (supplementary Fig.8C). To the authors' credit, they acknowledge this lack of correlation in the discussion.The authors have not provided any new data to substantiate their assertion that SNS-seq accurately detects origins in *T. brucei*, and therefore the work rests on a single experimental approach, without validation. As a result, the suggestion of abundant, previously undetected origins in the intergenic regions of multigene transcription remains a prediction. One key untested limitation of the work lies in the observation that the very large majority of SNS-seq signal overlaps with previously RNA-DNA hybrids; without an experimental test, the suggestion that the authors have 'disclosed for the first time a strong link between RNANA hybrid formation and DNA replication initiation' remains conjecture.
**Reviewer #2 (Public review):**
Summary:Stanojcic et al. investigate the origins of DNA replication in the unicellular parasite *Trypanosoma brucei*. They perform two experiments, stranded SNS-seq and DNA molecular combing. Further, they integrate various publicly available datasets, such as G4-seq and DRIP-seq, into their extensive analysis. Using this data, they elucidate the structure of origins of replications. In particular, they find various properties located at or around origins, such as polynucleotide stretches, G-quadruplex structures, regions of low and high nucleosome occupancy, R-loops, and that origins are mostly present in intergenic regions. Combining their population-level SNS-seq and their single-molecule DNA molecular combing data, they elucidate the total number of origins as well as the number of origins active in a single cell.Between the initial submission and this revision, the raised major concerns have not been resolved, and no additional validation has been provided.Strengths:(1) A very strong part of this manuscript is that the authors integrate several other datasets and investigate a large number of properties around origins of replication. Data analysis clearly shows the enrichment of various properties at the origins, and the manuscript is concluded with a very well-presented model that clearly explains the authors' understanding and interpretation of the data.(2) The DNA combing experiment is an excellent orthogonal approach to the SNS-seq data. The authors used the different properties of the two experiments (one giving location information, one giving single-molecule information) well to extract information and contrast the experiments.(3) The discussion is exemplary, as the authors openly discuss the strengths and weaknesses of the approaches used. Further, the discussion serves its purpose of putting the results in both an evolutionary and a trypanosome-focused context.Weaknesses:I have major concerns about the origin of replication sites determined from the SNS-seq data. As a caveat, I want to state that, before reading this manuscript, SNS-seq was unknown to me; hence, some of my concerns might be misplaced.(1) There are substantial discrepancies between the origins identified here and those reported in previous studies. Given that the other studies precede this manuscript, it is the authors' duty to investigate these differences. A conclusion should be reached on why the results are different, e.g., by orthogonally validating origins absent in the previous studies.

We agree that orthogonally validation of origins detected by stranded SNS-seq is necessary and we are working on it.

(2) I am concerned that up to 96% percent of all SNS-seq peaks are filtered away. If there is so much noise in the data, how can one be sure that the peaks that remain are real? Upon request, the authors have performed a control, where randomly placed peaks were run through the same filtering process. Only approximately twice as many experimental peaks passed filtering compared to random peaks. While the authors emphasize reproducibility between replicates, technical artifacts from the protocol would also be reproducible. Moreover, in other SNS-seq studies, for example, Pratto et al. Cell 2021, Fig. 1B, + and − strand peaks always appear closely paired. This pattern contrasts strongly with Fig. 2A in this manuscript.

The size and overlap of peaks depend on the length of the SNS. In our study, the width of the peaks corresponds to the size of the short nascent strands (0.5–2.5 kb) chosen as the starting material, whereas the width of the peaks in Pratto et al., Cell, 2021 are much larger (few kb). This could be due to the longer SNS used in the Pratto et al. study. Consequently, the overlap of the longer SNS is more pronounced since the SNS fibres elongate in both directions: at the 3′ end by DNA polymerase and at the 5′ end by ligation of Okazaki fragments. Additionally, the genomic regions displayed in our Figure 2A and in Pratto et al, Figure 1B are presented at substantially different resolutions, with a roughly ten‑fold difference in scale.

Further, I have some minor concerns that do not affect the main conclusions of the manuscript:- Fig 2C: The regions shown in the heatmap have different sizes, and I presume that the regions are ordered by size on the y-axis? If so, does the cone-shaped pattern, which is origin-less for genic regions and origin-enriched for intergenic regions, arise from the size of the regions? (I.e., for each genic region, the region itself is origin-less and the flanking intergenic regions contain origins.) If this is the case, then the peaks/valleys, centered exactly on the center of the regions on the mean frequency plots, arise from the different sizes of the analyzed regions, not from the fact that origins are mostly found at the center of intergenic regions. This data would be better presented with all regions stretched to the same size. This has not been addressed in the revision.

As the reviewer suggested, we have produced scaled plots of the stranded SNS-seq origins over genic and intergenic regions (see Figure 3, which is attached along with the Reviewer #2 (Recommendations for the authors)). However, we would prefer to keep the unscaled versions in the manuscript and add a note in the text as part of the Version of Record, explaining that the origins are evenly distributed throughout intergenic regions rather than being centred within them.

- Line 123, "and the average length of origins was found to be approximately 150 bp.": To determine origins, the authors filter away overlapping peaks and peaks that are too far from each other. Both restrict the minimal and maximal length of origins that can be observed, and this, in turn, affects the average length. This has not been addressed in the revision.

This observation is correct. By applying filtering and setting the maximum distance between the positive and negative peaks, we are most likely affecting the average length by excluding potentially wider origins.

We'll modify the text as part of the Version of Record.

Are claims well substantiated?:The identification of origins via SNS-seq appears to be incompletely supported to me.All downstream analyses depend on the reliability of origin identification.Impact:This study has the potential to be valuable for two fields: In research focused on *T. brucei* as a disease agent, where essential processes that function differently than in mammals are excellent drug targets. Further, this study would impact basic research analyzing DNA replication over the evolutionary tree, where *T. brucei* can be used as an early-divergent eucaryotic model organism.

The following is the authors’ response to the original reviews.

**eLife Assessment**
The authors use sequencing of nascent DNA (DNA linked to an RNA primer, "SNS-Seq") to localise DNA replication origins in *Trypanosoma brucei*, so this work will be of interest to those studying either Kinetoplastids or DNA replication. The paper presents the SNS-seq results for only part of the genome, and there are significant discrepancies between the SNS-Seq results and those from other, previously-published results obtained using other origin mapping methods. The reasons for the differences are unknown and from the data available, it is not possible to assess which origin-mapping method is most suitable for origin mapping in *T. brucei*. Thus at present, the evidence that origins are distributed as the authors claim - and not where previously mapped - is inadequate.

We would like to clarify a few points regarding our study. Our primary objective was to characterise the topology and genome-wide distribution of short nascent-strand (SNS) enrichments. The stranded SNS-seq approach provides the high strand-specific resolution required to analyse origins. The observation that SNS-seq peaks (potential origins) are most frequently found in intergenic regions is not an artefact of analysing only part of the genome; rather, it is a result of analysing the entire genome.

We agree that orthogonal validation is necessary. However, neither MFA-seq nor TbORC1/CDC6 ChIP-on-chip has yet been experimentally validated as definitive markers of origin activity in *T. brucei*, nor do they validate each other.

**Public Reviews:**

**Reviewer #1 (Public review):**
In this paper, Stanojcic and colleagues attempt to map sites of DNA replication initiation in the genome of the African trypanosome, *Trypanosoma brucei*. Their approach to this mapping is to isolate 'short-nascent strands' (SNSs), a strategy adopted previously in other eukaryotes (including in the related parasite Leishmania major), which involves isolation of DNA molecules whose termini contain replication-priming RNA. By mapping the isolated and sequenced SNSs to the genome (SNS-seq), the authors suggest that they have identified origins, which they localise to intergenic (strictly, inter-CDS) regions within polycistronic transcription units and suggest display very extensive overlap with previously mapped R-loops in the same loci. Finally, having defined locations of SNS-seq mapping, they suggest they have identified G4 and nucleosome features of origins, again using previously generated data.Though there is merit in applying a new approach to understand DNA replication initiation in *T. brucei*, where previous work has used MFA-seq and ChIP of a subunit of the Origin Replication Complex (ORC), there are two significant deficiencies in the study that must be addressed to ensure rigour and accuracy.(1) The suggestion that the SNS-seq data is mapping DNA replication origins that are present in inter-CDS regions of the polycistronic transcription units of *T. brucei* is novel and does not agree with existing data on the localisation of ORC1/CDC6, and it is very unclear if it agrees with previous mapping of DNA replication by MFA-seq due to the way the authors have presented this correlation. For these reasons, the findings essentially rely on a single experimental approach, which must be further tested to ensure SNS-seq is truly detecting origins. Indeed, in this regard, the very extensive overlap of SNS-seq signal with RNA-DNA hybrids should be tested further to rule out the possibility that the approach is mapping these structures and not origins.(2) The authors' presentation of their SNS-seq data is too limited and therefore potentially provides a misleading view of DNA replication in the genome of *T. brucei*. The work is presented through a narrow focus on SNS-seq signal in the inter-CDS regions within polycistronic transcription units, which constitute only part of the genome, ignoring both the transcription start and stop sites at the ends of the units and the large subtelomeres, which are mainly transcriptionally silent. The authors must present a fuller and more balanced view of SNS-seq mapping across the whole genome to ensure full understanding and clarity.

Regarding comparisons with previous work:

- Two other attempts to identify origins in *T. brucei* - ORC1/CDC6 binding sites (ChIP-on-chip, PMID: 22840408) and MFA-seq (PMID: 22840408, 27228154) - were both produced by the McCulloch group. These methods do not validate each other; in fact, MFA-seq origins overlap with only 4.4% of the 953 ORC1/CDC6 sites (PMID: 29491738). Therefore, low overlap between SNS-seq peaks and ORC1/CDC6 sites cannot disqualify our findings. Similar low overlaps are observed in other parasites (PMID: 38441981, PMID: 38038269, PMID: 36808528) and in human cells (PMID: 38567819).

- We also would like to emphasize that the ORC1/CDC6 dataset originally published (PMID: 22840408) is no longer available; only a re-analysis by TritrypDB exists, which differs significantly from the published version (personal communication from Richard McCulloch). While the McCulloch group reported a predominant localization of ORC1/CDC6 sites within SSRs at transcription start and termination regions, our re-analysis indicates that only 10.3% of TbORC1/CDC6-12Myc sites overlapped with 41.8% of SSRs.

- MFA-seq does not map individual origins, it rather detects replicated genomic regions by comparing DNA copy number between S- and G1-phases of the cell cycle (PMID: 36640769; PMID: 37469113; PMID: 36455525). The broad replicated regions (0.1–0.5 Mbp) identified by MFA-seq in *T. brucei* are likely to contain multiple origins, rather than just one. In that sense we disagree with the McCulloch's group who claimed that there is a single origin per broad peak. Our analysis shows that up to 50% of the origins detected by stranded SNS-seq locate within broad MFA-seq regions. The methodology used by McCulloch’s group to infer single origins from MFA-seq regions has not been published or made available, as well as the precise position of these regions, making direct comparison difficult.

Finally, the genomic features we describe—poly(dA/dT) stretches, G4 structures and nucleosome occupancy patterns—are consistent with origin topology described in other organisms.

On the concern that SNS-seq may map RNA-DNA hybrids rather than replication origins: Isolation and sequencing of short nascent strands (SNS) is a well-established and widely used technique for high-resolution origin mapping. This technique has been employed for decades in various laboratories, with numerous publications documenting its use. We followed the published protocol for SNS isolation (Cayrou et al., Methods, 2012, PMID: 22796403). RNA-DNA hybrids cannot persist through the multiple denaturation steps in our workflow, as they melt at 95°C (Roberts and Crothers, *Science*, 1992; PMID: 1279808). Even in the unlikely event that some hybrids remained, they would not be incorporated into libraries prepared using a single-stranded DNA protocol and therefore would not be sequenced (see Figure 1B and Methods).

Furthermore, our analysis shows that only a small proportion (1.7%) of previously reported RNA-DNA hybrids overlap with SNS-seq origins. It is important to note that RNA-primed nascent strands naturally form RNA-DNA hybrids during replication initiation, meaning the enrichment of RNA-DNA hybrids near origins is both expected and biologically relevant.

On the claim that our analysis focuses narrowly on inter-CDS regions and ignores other genomic compartments: this is incorrect. We mapped and analyzed stranded SNS-seq data across the entire genome of *T. brucei* 427 wild-type strain (Müller et al., Nature, 2018; PMID: 30333624), including both core and subtelomeric regions. Our findings indicate that most origins are located in intergenic regions, but all analyses were performed using the full set of detected origins, regardless of location.

We did not ignore transcription start and stop sites (TSS/TTS). The manuscript already includes origin distribution across genomic compartments as defined by TriTrypDB (Fig. 2C) and addresses overlap with TSS, TTS and HT in the section “Spatial coordination between the activity of the origin and transcription”. While this overlap is minimal, we have included metaplots in the revised manuscript for clarity.

**Reviewer #2 (Public review):**
Summary:Stanojcic et al. investigate the origins of DNA replication in the unicellular parasite *Trypanosoma brucei*. They perform two experiments, stranded SNS-seq and DNA molecular combing. Further, they integrate various publicly available datasets, such as G4-seq and DRIP-seq, into their extensive analysis. Using this data, they elucidate the structure of the origins of replication. In particular, they find various properties located at or around origins, such as polynucleotide stretches, G-quadruplex structures, regions of low and high nucleosome occupancy, R-loops, and that origins are mostly present in intergenic regions. Combining their population-level SNS-seq and their single-molecule DNA molecular combing data, they elucidate the total number of origins as well as the number of origins active in a single cell.Strengths:(1) A very strong part of this manuscript is that the authors integrate several other datasets and investigate a large number of properties around origins of replication. Data analysis clearly shows the enrichment of various properties at the origins, and the manuscript concludes with a very well-presented model that clearly explains the authors' understanding and interpretation of the data.

We sincerely thank you for this positive feedback.

(2) The DNA combing experiment is an excellent orthogonal approach to the SNS-seq data. The authors used the different properties of the two experiments (one giving location information, one giving single-molecule information) well to extract information and contrast the experiments.

Thank you very much for this remark.

(3) The discussion is exemplary, as the authors openly discuss the strengths and weaknesses of the approaches used. Further, the discussion serves its purpose of putting the results in both an evolutionary and a trypanosome-focused context.

Thank you for appreciating our discussion.

Weaknesses:I have major concerns about the origin of replication sites determined from the SNS-seq data. As a caveat, I want to state that, before reading this manuscript, SNS-seq was unknown to me; hence, some of my concerns might be misplaced.(1) I do not understand why SNS-seq would create peaks. Replication should originate in one locus, then move outward in both directions until the replication fork moving outward from another origin is encountered. Hence, in an asynchronous population average measurement, I would expect SNS data to be broad regions of + and -, which, taken together, cover the whole genome. Why are there so many regions not covered at all by reads, and why are there such narrow peaks?

Thank you for asking these questions. As you correctly point out, replication forks progress in both directions from their origins and ultimately converge at termination sites. However, the SNS-seq method specifically isolates short nascent strands (SNSs) of 0.5–2.5 kb using a sucrose gradient. These short fragments are generated immediately after origin firing and mark the sites of replication initiation, rather than the entire replicated regions. Consequently: (i) SNS-seq does not capture long replication forks or termination regions, only the immediate vicinity of origins. (ii) The narrow peaks indicate the size of selected SNSs (0.5–2.5 kb) and the fact that many cells initiate replication at the same genomic sites, leading to localized enrichment. (iii) Regions without coverage refer to genomic areas that do not serve as efficient origins in the analyzed cell population. Thus, SNS-seq is designed to map origin positions, but not the entire replicated regions.

(2) I am concerned that up to 96% percent of all peaks are filtered away. If there is so much noise in the data, how can one be sure that the peaks that remain are real? Specifically, if the authors placed the same number of peaks as was measured randomly in intergenic regions, would 4% of these peaks pass the filtering process by chance?

Maintaining the strandness of the sequenced DNA fibres enabled us to filter the peaks, thereby increasing the probability that the filtered peak pairs corresponded to origins. Two SNS peaks must be oriented in a way that reflects the topology of the SNS strands within an active origin: the upstream peak must be on the minus strand and followed by the downstream peak on the plus strand.

As suggested by the reviewer, we tested whether randomly placed plus and minus peaks could reproduce the number of filter-passing peaks using the same bioinformatics workflow. Only 1–6% of random peaks passed the filters, compared with 4–12% in our experimental data, resulting in about 50% fewer selected regions (origins). Moreover, the “origins” from random peaks showed 0% reproducibility across replicates, whereas the experimental data showed 7–64% reproducibility. These results indicate that the retainee peaks are highly unlikely to arise by chance and support the specificity of our approach. Thank you for this suggestion.

(3) There are 3 previous studies that map origins of replication in *T. brucei*. Devlin et al. 2016, Tiengwe et al. 2012, and Krasiļņikova et al. 2025 (https://doi.org/10.1038/s41467-025-56087-3), all with a different technique: MFA-seq. All three previous studies mostly agree on the locations and number of origins. The authors compared their results to the first two, but not the last study; they found that their results are vastly different from the previous studies (see Supplementary Figure 8A). In their discussion, the authors defend this discrepancy mostly by stating that the discrepancy between these methods has been observed in other organisms. I believe that, given the situation that the other studies precede this manuscript, it is the authors' duty to investigate the differences more than by merely pointing to other organisms. A conclusion should be reached on why the results are different, e.g., by orthogonally validating origins absent in the previous studies.

The MFA-seq data for *T. brucei* were published in two studies by McCulloch’s group: Tiengwe et al. (2012) using TREU927 PCF cells, and Devlin et al. (2016) using PCF and BSF Lister427 cells. In Krasilnikova et al. (2025), previously published MFA-seq data from Devlin et al. were remapped to a new genome assembly without generating new MFA-seq data, which explains why we did not include that comparison.

Clarifying the differences between MFA-seq and our stranded SNS-seq data is essential. MFA-seq and SNS-seq interrogate different aspects of replication. SNS-seq is a widely used, high-resolution method for mapping individual replication origins, whereas MFA-seq detects replicated regions by comparing DNA copy number between S and G1 phases. MFA-seq identified broad replicated regions (0.1–0.5 Mb) that were interpreted by McCulloch’s group as containing a single origin. We disagree with this interpretation and consider that there are multiple origins in each broad peaks; theoretical considerations of replication timing indicate that far more origins are required for complete genome duplication during the short S-phase. Once this assumption is reconsidered, MFA-seq and SNS-seq results become complementary: MFA-seq identifies replicated regions, while SNS-seq pinpoints individual origins within those regions. Our analysis revealed that up to 50% of the origins detected by stranded SNS-seq were located within the broad MFA peaks. This pattern—broad MFA-seq regions containing multiple initiation sites—has also recently been found in *Leishmania* by McCulloch’s team using nanopore sequencing (PMID: 26481451). Nanopore sequencing showed numerous initiation sites within MFA-seq regions and additional numerous sites outside these regions in asynchronous cells, consistent with what we observed using stranded SNS-seq in *T. brucei*. We will expand our discussion and conclude that the discrepancy arises from methodological differences and interpretation. The two approaches provide complementary insights into replication dynamics, rather than ‘vastly different’ results.

We recognize the importance of validating our results in future using an alternative mapping method and functional assays. However, it is important to emphasize that stranded SNS-seq is an origin mapping technique with a very high level of resolution. This technique can detect regions between two divergent SNS peaks, which should represent regions of DNA replication initiation. At present, no alternative technique has been developed that can match this level of resolution.

(4) Some patterns that were identified to be associated with origins of replication, such as G-quadruplexes and nucleosomes phasing, are known to be biases of SNS-seq (see Foulk et al. Characterizing and controlling intrinsic biases of lambda exonuclease in nascent strand sequencing reveals phasing between nucleosomes and G-quadruplex motifs around a subset of human replication origins. Genome Res. 2015;25(5):725-735. doi:10.1101/gr.183848.114).

It is important to note that the conditions used in our study differ significantly from those applied in the Foulk et al. Genome Res. 2015. We used SNS isolation and enzymatic treatments as described in previous reports (Cayrou, C. et al. Genome Res, 2015 and Cayrou, C et al. Methods, 2012). Here, we enriched the SNS by size on a sucrose gradient and then treated this SNS-enriched fraction with high amounts of repeated λ-exonuclease treatments (100u for 16h at 37oC - see Methods). In contrast, Foulk et al. used sonicated total genomic DNA for origin mapping, without enrichment of SNS on a sucrose gradient as we did, and then they performed a λ-exonuclease treatment. A previous study (Cayrou, C. et al. Genome Res, 2015, Figure S2, which can be found at https://genome.cshlp.org/content/25/12/1873/suppl/DC1) has shown that complete digestion of G4-rich DNA sequences is achieved under the conditions we used.

Furthermore, the SNS depleted control (without RNA) was included in our experimental approach. This control represents all molecules that are difficult to digest with lambda exonuclease, including G4 structures. Peak calling was performed against this background control, with the aim of removing false positive peaks resulting from undigested DNA structures. We explained better this step in the revised manuscript.

The key benefit of our study is that the orientation of the enrichments (peaks) remains consistent throughout the sequencing process. We identified an enrichment of two divergent strands synthesised on complementary strands containing G4s. These two divergent strands themselves do not, however, contain G4s (see Fig. 8 for the model). Therefore, the enriched molecules detected in our study do not contain G4s. They are complementary to the strands enriched with G4s. This means that the observed enrichment of

G4s cannot be an artefact of the enzymatic treatments used in this study. We added this part in the discussion of the revised manuscript.

We also performed an additional control which is not mentioned in the manuscript. In parallel with replicating cells, we isolated the DNA from the stationary phase of growth, which primarily contains non-replicating cells. Following the three λ-exonuclease treatments, there was insufficient DNA remaining from the stationary phase cells to prepare the libraries for sequencing. This control strongly indicated that there was little to no contaminating DNA present with the SNS molecules after λ-exonuclease enrichment.

**Recommendations for the authors:**

**Reviewer #1 (Recommendations for the authors):**
Four broad issues need to be addressed.(1) The authors have attempted to test the overlap between ORC1/CDC6 (an ORC subunit) binding in the genome and SNS-seq. If there were an overlap, this would provide evidence that the SNS-seq signals represent origins. However, the analysis provided is inadequate: merely a statement that "we obtained an overlap of 4.2% between origins and ORC1/CDC6 binding sites within a window of {plus minus}2 kb and 6.2% in the window of {plus minus}3 kb". Nowhere are these data shown or properly discussed:a) The authors need to provide a diagram showing where in the genome the very small amount of overlapping SNS-seq and ORC1/CDC6 binding occurs, and to clearly show and state how many of the intergenic SNS-seq peaks are sites of ORC1/CDC6 binding. In the absence of such analysis, a key question is unanswered: is there any evidence of ORC1/CDC6 (or ORC more broadly) binding at the SNS-seq signals within the polycistronic transcription units?

In the original version of the manuscript, these data were already presented as percentages in the text and as a metaplot (Supplementary Fig. 8C).

We based our analysis on the set of 350 TbORC1/CDC6 binding sites available on TriTrypDB at the time of analysis. This dataset was a filtered subset of the originally reported TbORC1/CDC6 ChIP‑on‑chip peaks (personal communication, TriTrypDB). Since then, the unfiltered dataset has been made available. We therefore re‑analyzed the overlap using this dataset, to which we applied a filtering that yielded 990 binding sites closely matching the 953 sites reported by the McCulloch group. We need to stress here that the original 953 sites reported by the McCulloch group (Tiengwe et al., 2012 PMID: 22840408), is not available anymore and that the authors:

- do not provide genomic coordinates for the 953 binding sites and

- do not release any scripts or methodology that would allow independent reproduction of the 953 sites.

A similar remark also applies to the MFA-seq data (see below).

To address the reviewer’s request, we have now:

(1) Recalculated the overlap using the updated TbORC1/CDC6 dataset (990 binding sites) from TriTrypDB.

(2) Added the absolute number of overlapping SNS‑seq origins and TbORC1/CDC6 binding sites in the Results section for clarity.

(3) Included the TbORC1/CDC6 binding sites in the chromosomal overview (newly added to Supplementary Fig. 8A), so that their genomic localization relative to SNS‑seq peaks is visually accessible.

(4) Revised the metaplots of TbORC1/CDC6 distribution around SNS‑seq origins using the updated dataset (Supplementary Fig. 8C).

With these improvements, we now find that:

- Within ±2 kb, 12.9% (253) of SNS‑seq origins overlap with 25.6% of TbORC1/CDC6 binding sites.

- Within ±3 kb, 18.8% (370) of SNS‑seq origins overlap with 37.4% of TbORC1/CDC6 binding sites.

The updated metaplot shows a clear depletion of TbORC1/CDC6 signal at the origin center, with modest enrichment ~5 kb upstream and downstream. The underlying reason for this pattern remains unknown, and we agree that additional studies will be needed to understand it.

b) Equally, the authors need to explain what they conclude from this analysis. They make a comparison with T. cruzi ORC1/CDC6 and SNS-seq overlap, which does not illuminate what the data tell us. For instance, if there is no or minimal overlap between ORC1/CDC6 binding and SNS-seq peaks within the polycistronic transcription units, do they conclude that the major SNS-seq signal they detail is evidence for ORC-independent DNA replication? If there is no overlap, what further evidence can they provide that these signals truly are origins?

First, we would like to clarify that, to date, there is no evidence supporting ORC‑independent DNA replication in *T. brucei*, and—importantly—no published data demonstrating that TbORC1/CDC6 is universally required for DNA replication initiation. Because of this, we consider that it would be inappropriate to conclude that regions lacking detectable TbORC1/CDC6 signal undergo ORC‑independent initiation. We would prefer not to speculate in the absence of supporting evidence and would gratefully consider any reference the reviewer wishes to provide on this subject.

Second, the low overlap between TbORC1/CDC6 binding sites and SNS‑seq origins does not, in our view, invalidate our mapping of replication initiation sites. Multiple factors contribute to this:

(1) Low overlap between ORC1/CDC6 and origin‑mapping techniques has been repeatedly reported across kinetoplastids. For instance, in *T. cruzi*, 88.2% of origins detected by DNAscent nanopore sequencing showed no overlap with TcORC1/CDC6–Ty1 ChIP signal within ±3 kb, and only 11.7% co‑localized. This is strikingly similar to our observations in *T. brucei.* Thus, our data are consistent with the broader pattern in trypanosomatids rather than an exception.

(2) The origin topology detected by stranded SNS‑seq is supported by several genomic characteristic found frequently in other eukaryotes, including:

- A highly specific and polarized poly(dA)/poly(dT) sequence environment.

- Strand‑specific G4 structures positioned around origin centers.

- A conserved nucleosome‑depleted region flanked by well‑positioned nucleosomes.

These features are absent from shuffled controls, appear at high significance, and recapitulate hallmark signatures of replication origins in other eukaryotes.

Together, these findings give us confidence that the SNS‑seq peaks represent genuine origins - despite the incomplete overlap with TbORC1/CDC6 binding.

Third, we fully agree with the reviewer that a definitive conclusion would require an additional, independent validation method.

Given the lack of complete ORC subunit datasets and the unusual biology of trypanosomatid replication complexes, we believe that the cautious interpretation above is the most appropriate.

c) The authors state (Discussion): "Validation of origins is generally a difficult task, particularly in trypanosomatids, where proteins involved in the initiation of DNA replication are difficult to determine. Few proteins have been described as potential ORC subunits (reviewed in 61), and none of them have been shown to be a specific marker that indicates the origins." There are two problems with the statement. First, most of the subunits of ORC have now been described in *T. brucei*; the authors should make this clear. Second, mapping of ORC1/CDC6 localisation, contrary to what the authors state here, shows precise correlation with the peaks of every MFA-seq signal described (see Tiengwe et al, Cell Reports, 2012); thus, ORC1/CDC6 binding provides evidence that MFA-seq is detecting origins, something that cannot be said for SNS-seq. The authors need to correct this misleading paragraph.

As suggested, we have removed the paragraph from the Discussion to avoid confusion. However, we disagree with the reviewer's assessment and clarify below our position regarding the issues raised.

First, we agree that five candidate ORC subunits have now been identified in *T. brucei*. Our intention was not to suggest the contrary, but rather to emphasize that, although candidate ORC components have been described, direct functional evidence for their roles in replication initiation is still limited. For this reason, we were cautious in referring to any ORC component as a definitive marker of replication origins.

Second, regarding the reviewer’s statement that TbORC1/CDC6 binding “shows precise correlation with the peaks of every MFA‑seq signal”, we respectfully disagree based on several observations:

(1) MFA‑seq does not identify individual origin centers, but rather broad replicated regions that often span hundreds of kilobases. By design, this method cannot define the number or position of discrete origins within each peak. For that reason, MFA-seq regions do not have the resolution required to validate TbORC1/CDC6 binding sites as individual origins.

(2) In the published datasets (Tiengwe et al., Devlin et al.), no metaplots or locus‑wide quantification of the overlap between MFA‑seq peaks and TbORC1/CDC6 binding were provided. The coordinates or the approach used to define the discrete regions that they define as the originsin the MFA‑seq broad peaks have never been described or made available, making it difficult to evaluate the claimed correspondence.

(3) Notably, McCulloch’s group later reported that only 4.4% of the 953 TbORC1/CDC6 sites overlapped with their 42 MFA‑seq “origins”, underscoring that the degree of correspondence is in fact limited (PMID: 29491738).

(4) Finally, as noted in our response to point (1b), low overlap between ORC1/CDC6 binding sites and origin‑mapping techniques is a consistent observation across kinetoplastids, including *T. cruzi*, where DNAscent‑mapped origins show only ~12% overlap with TcORC1/CDC6 ChIP signals. This suggests that the limited overlap we observe is not unique to our dataset.

For these reasons, we are not convinced that the TbORC1/CDC6 binding sites have been shown to align precisely with MFA seq peaks, nor that these datasets definitively validate origin mapping in *T. brucei*. Nevertheless, to avoid over‑interpretation and potential confusion, we have removed the paragraph from the Discussion as requested. We hope this clarifies our position and improves the accuracy and neutrality of the manuscript.

(2) Like for ORC1/CDC6 localisation, the authors' evaluation of the relationship between MFA-seq and SNS-seq mapping is inadequate, and the depth of the analysis and discussion needs to be improved:a) The authors state: "We found 28-42% stranded SNS-seq origins overlapped with early and 43-55% overlapped with late S-phase MFA-seq replicated regions (Supplementary Figure 8B)." This seems important and provides (limited) validation of both datasets, but cannot be discerned from the supplied figure. Please provide a metaplot of the two datasets centred on the MFA-seq loci, including the SNS-seq peak amplitude.

We would like to emphasize that MFA‑seq is not a method designed to map individual origins, and this fundamentally limits the interpretability of metaplots centered on MFA-seq regions. MFA‑seq identifies broad replication‑enriched domains, typically spanning 100–500 kb, within which multiple origins may fire asynchronously across the cell population.

This concern is reinforced by the original MFA‑seq publications (Tiengwe et al., 2012; Devlin et al., 2016), which:

- do not provide positional data for the 42-47 MFA‑inferred origins,

- do not describe the computational method used to derive individual origin coordinates from the broad peaks, and

- do not release any scripts or methodology that would allow independent reproduction of the claimed origin positions.

Because of this, it is not possible to reconstruct or validate how the 42 MFA‑seq “origin” sites were defined, nor to use those coordinates as anchors for metaplot analyses.

Most importantly, we disagree with the underlying assumption that each MFA‑seq peak corresponds to exactly one origin. This assumption runs counter to the principle of the technique, which identifies regions of higher DNA content in replicating cells than in non-replicating cells; it is also contradicted by our stranded SNS‑seq data and by DNA combing measurements:

- SNS‑seq detects multiple discrete origins within the same genomic regions that produce a single broad MFA‑seq peak.

- DNA combing reveals inter‑origin distances of ~36–422 kb (median ~150 kb) (PMID: 26976742), which is far shorter than the ~400–600 kb replication domains identified by MFA‑seq.

- Furthermore, with only 42 origins detected by MFA-seq, it is not possible to achieve complete genome replication in *T. brucei* during S-phase. DNA combing has found that the average speed of replication forks in the procyclic forms is 1.9 Kb/min. (PMID: 26976742). Dividing the size of the *Trypanosoma brucei brucei* TREU927 genome (26.1 Mb) by 42 origins (PMID: 22840408) shows that 621 Kb must be replicated during the S phase. Using the calculated average replication speed of 1.9 Kb/min, we can estimate that the replication of 621 Kb would take 327 min (5.45 hours) (621 Kb/1.9 Kb/min = 327 min). However, this exceeds the estimated length of the S-phase in these parasites, which is 2.31 hours (138.6 minutes) (PMID: 32397111, 31811174, 28258618) or less, 1.36 hours (PMID: 2190996, 10574712) in *Trypanosoma brucei* procyclic forms. Therefore, more than 42 origins are necessary to complete replication during the short S phase.

This makes it unlikely that MFA-seq regions represent single functional origins. For these reasons, a metaplot centered on MFA‑seq “loci” may lead to misinterpretations and would not provide biologically meaningful information.

We hope that the expanded explanation clarifies our interpretation of the relationship between these two complementary, but fundamentally different, methods.

b) The authors state that "Our results showed that the origins are predominantly located in the intergenic regions within the PTUs (Figure 2C)'. This finding cannot be discerned from this figure, which does not show 'strand switch regions' (SSRs; transcription start/stop sites), where MFA-seq predicts all origins to localise. The authors need to acknowledge this difference and must show a comparison of SNS-seq data, including peak amplitude, around all SSRs (whether predicted by MFA-seq to act as origins or not, since all appear to bind ORC1/CDC6).

We have now provided the metaplots showing the overlap between stranded SNS-seq origins and SSRs (see Supplementary Figure 8D). This difference has been acknowledged and discussed in the revised manuscript.

c) Finally, the authors' interpretation that around 30-55% of SNS-seq peaks overlap with MFA-seq 'origins' is highly questionable. MFA-seq peaks are regions of increased DNA content in replicating cells relative to non-replicating cells, and so the entire region under the MFA-seq peak is not necessarily an origin, but is likely to be a more discrete locus (eg, the SSR, where ORC1/CDC6 mainly localises). They should correct the wording and discuss what significance they see in this overlap; for instance, do they think SNS-seq 'clusters' are more pronounced within the MFA-seq peaks and, if so, what might this mean, and why does it not correlate with ORC1/CDC6 localisation?

As the reviewer notes, ‘MFA‑seq peaks are regions of increased DNA content, and so the entire region under the MFA-seq peak is not necessarily an origin but is likely to be a more discrete locus’. This is exactly why MFA‑seq is inappropriate for identifying discrete/individual origins: within these replicated domains, multiple origins can fire, as revealed both by stranded SNS‑seq mapping.

Regarding the overlap between SNS‑seq origins and MFA‑seq peaks, we agree with the reviewer that this overlap should not be interpreted as validating MFA‑seq “origin positions.” Instead, we now describe it more accurately as the proportion of discrete SNS‑seq origins that fall within broader MFA‑seq replication domains. This is expected, because SNS‑seq identifies individual initiation events, whereas MFA‑seq identifies S‑phase replication domains averaged across a population. Our stranded SNS‑seq data do not show enhanced origin accumulation within MFA-seq regions, and we find no correlation with TbORC1/CDC6 positions. This is now discussed.

Regarding SSRs, we do not share the view that they should be considered privileged initiation sites. After remapping the TbORC1/CDC6 ChIP‑on‑chip dataset (see above) to the *T. brucei* Lister 427–2018 genome (Supplementary Fig. 8A), we observed that TbORC1/CDC6 binding is distributed throughout the chromosomes, not restricted to SSRs. To quantify this, we analyzed the overlap between TbORC1/CDC6 sites and all annotated SSR classes (dSSRs, cSSRs, and head‑to‑tail regions, as defined in Kim et al. 2009). The results show that:

Only 10% of TbORC1/CDC6 binding sites fall within 40% of all SSRs.

At the level of individual SSR types:

- TTS: 3.3% of TTS overlap with 0.3% of TbORC1/CDC6 sites.

- TSS: 67% of TSS overlap with 6.1% of TbORC1/CDC6 sites.

- Head‑to‑tail regions: 54.2% overlap with 3.6% of TbORC1/CDC6 sites.

These analyses demonstrate that most TbORC1/CDC6 sites are *not* located at SSRs, contradicting the idea that SSRs represent primary or exclusive origin sites.

**Author response image 1. sa3fig1:** Overlap between TbORC1/CDC6-12Myc binding sites (Tiengwe 2012, Cell Reports) and strand‑switch regions (SSRs). Venn diagram showing the overlap of 990TbORC1/CDC6-12Mycbinding sites (Retrieved from TritrypDB filtered at score 22 to achieve a number of binding sites similar to the one (953 binding sites) published in Tiengwe 2012, Cell Reports) and SSR sites in the genome (Kim 2018, NAR). The intersection shows that 10.3% of Orc1/CDC6 binding sites overlap with 41.8% SSRs. The intersection is subdivided into TSS (orange), TTS in (blue) and HT in (green).

(3) A key objection to the data presentation is the decision to limit SNS-seq mapping to the intergenic regions. In addition to overlooking the SSRs (see above, 2), so-called subtelomeres, which account for nearly 50% of the *T. brucei* genome and are largely untranscribed, are not shown or discussed at all. Providing this data will improve clarity and also provide a key test of one of the predictions that the authors make: "most origins are localized in actively transcribed regions, which could lead to collisions between DNA replication and the transcription machinery. This spatial coincidence implies that transcription and replication must occur in a highly ordered and cooperative manner in *T. brucei*."

We do not understand why this reviewer concluded that we took 'the decision to limit the mapping of SNS-seq to intergenic regions'. This is a factual error.

To be clearer,

(2) We now explicitly present the distribution of SNS‑seq origins across core and subtelomeric regions in the revised Figure 2D, making clear that origin mapping was performed genome‑wide.

(2) And that SNS‑seq origins are also present in subtelomeric regions. We have revised the manuscript to avoid any implication that origin firing is restricted only to actively transcribed regions. Our data show that most SNS‑seq origins lie within intergenic regions of PTUs, but a minority are found outside these regions—including subtelomeres and SSRs. The revised text reflects this nuance and highlights that the spatial relationship between transcription and replication is strong but not exclusive.

These additions undoubtedly ensure that the genomic-wide nature of SNS-seq analysis is transparent to the reader and should therefore remove this reviewer's “key objection”.

a) The authors must show SNS-seq mapping to the subtelomeres (in addition to around the SSRs; see comment (2). If no SNS-seq peaks are detected in the subtelomeres, what do the authors conclude about how the genome is duplicated? If SNS-seq peaks are detected in the subtelomeres, do they correspond with the ordered nucleosomes in this part of the genome described by Maree et al (PMID: 28344657); if so, might SNS-seq signal localisation not be directed by transcription but chromatin?

We have now presented the proportion of origins in subtelomeric regions (see Figure 2B).

As illustrated in the metaplots in Author response image 2, the distribution of nucleosomes around the subtelomeric origins is similar to the distribution shown for all origins in the manuscript. We do not see the pattern of nucleosomes as described by Maree et al (PMID: 28344657) over ORC1/CDC6 binding sites in this part of the genome.

**Author response image 2. sa3fig2:** Metaplots showing the mean nuclesome signal over centred SNS-seq origins in subtelomeric regions. Two replicates from Maree et al 2019 (PMID: 28344657).

We never claimed that transcription directs the localisation of the SNS-seq signal. We did not conduct experiments to address this issue. In contrast, we consider that the organisation of chromatin exerts a significant influence on the selection of active origins.

(4) The major conclusion of the manuscript is that the SNS-seq signal corresponds very precisely to the locations of RNA-DNA hybrids (R-loops). Given all the limitations discussed above, can the authors rule out the possibility that SNS-seq is merely mapping DNA-DNA hybrids and is not, in fact, detecting origins?a) It is legitimate to speculate about the possibility that the very extensive overlap between SNS-seq and DRIP-seq signals within polycistronic transcription units (between ORFs) might suggest that DRIP-seq data detects nascent strands at replication origins, rather than R-loops at sites of pre-mRNA processing, as previously suggested by Briggs et al (PMID: 30304482). (eg, 'we disclosed for the first time a strong link between R-loop formation and DNA replication initiation'; 'The RNA:DNA hybrids are formed at initiation sites by RNA priming of SNS and Okazaki fragments'). However, the authors should acknowledge that alternative explanations for the localisation and potential functions of inter-CDS R-loops have been suggested,

We do not find extensive overlap between stranded SNS-seq and DRIP-seq signal. We have observed only a minor proportion (1.7%) of the previously reported DRIP-seq signal to overlap with the origins detected by stranded SNS-seq. The RNA-primed SNS must form RNA:DNA hybrids during the initiation of DNA replication, and that an enrichment of these hybrids around the origins is expected. Therefore, we legitimately speculated that this minor proportion of RNA:DNA hybrids enriched around origin centres could be due to the origin activation.

We agree that some of the DRIP-seq signals detected around the origins may be sites of pre-mRNA processing, as previously suggested by Briggs et al. (PMID: 30304482). Since there is no data proving implication of pre-mRNA processing into DNA replication initiation we prefer not to speculate about it.

b) More importantly, the authors should provide experimental evidence that tests such a mechanistic prediction of R-loops and origins: for instance, have they attempted to remove R-loops, eg, by treatment with RNase H, and checked that the SNS-seq signal is unaltered? In the absence of such data, they cannot exclude the possibility that their work has revealed an overlooked problem with SNS-seq (which may not be limited to *T. brucei*; are matched DRIP-seq and SNS-seq datasets available to correlate these signals in a range of organisms?).

We have not attempted RNase H treatment for a fundamental methodological reason: it seems highly improbable that RNA:DNA hybrids would persist through the multiple denaturation steps inherent to the SNS‑seq enrichment protocol. Published biophysical measurements show that RNA:DNA hybrids melt at ~95 °C (Roberts & Crothers, *Science*, 1992; PMID: 1279808), which is the temperature repeatedly applied during SNS isolation. Under these conditions, persistent RNA:DNA hybrids cannot remain intact and therefore cannot be responsible for the SNS‑seq peaks detected.

We do not interpret our findings as revealing an “overlooked problem with SNS‑seq.” Instead, we consider that the enrichment of RNA:DNA hybrids around origins observed in DRIP‑seq is biologically meaningful and expected, given that replication initiation involves RNA‑primed nascent strands and that DRIP‑seq detects such structures.

**Reviewer #2 (Recommendations for the authors):**
I have some minor concerns that do not affect the main conclusions of the manuscript:(1) Figure 2B: The regions shown in the heatmap have different sizes, and I presume that the regions are ordered by size on the y-axis? If so, does the cone-shaped pattern, which is origin-less for genic regions and origin-enriched for intergenic regions, arise from the size of the regions? (I.e., for each genic region, the region itself is origin-less and the flanking intergenic regions contain origins.) If this is the case, then the peaks/valleys, centered exactly on the center of the regions on the mean frequency plots, arise from the different sizes of the analyzed regions, not from the fact that origins are mostly found at the center of intergenic regions.

That is correct. The regions displayed in the heatmaps are genic and intergenic region sorted by size. We did not want to convey with this metaplot that the origins are accumulating at the centres of the intergenic region but mainly that genic regions are mostly devoid of origins and the intergenic regions enriched in origins.

(2) Line 123, "and the average length of origins was found to be approximately 150 bp.": To determine origins, the authors filter away overlapping peaks and peaks that are too far from each other. Both restrict the minimal and maximal length of origins that can be observed, and this, in turn, affects the average length.

This observation is correct. By applying filtering and setting the maximum distance between the positive and negative peaks, we are most likely affecting the average length by excluding origins that are potentially wider. Nevertheless, the violin plot shows that the majority of origins are shorter than 500 nt. In the end, the size of regions detected as the origin is not important. What gives the resolution of stranded-SNS-seq is the ability to identify the centre of the origin between the minus and plus peaks.

(3) Data in the manuscript were sometimes not presented in an easy-to-read manner. In some cases, this was due to benign things, such as missing labels for the mean frequency plots (e.g., Figure 2B, blue and green) or very small fonts for axes (Figure 2B). Sometimes, due to the plot types that were chosen, such as pie-charts (Figure 2C, see here), stacked bar plots (Figure 6B), or showing cumulative distributions (Figure 5C, and Figure 2D) it makes it difficult to judge the actual distribution.

Wherever possible, the size of the small fonts was increased to the maximum. Missing labels were added to the mean frequency plots. We increased the font size for the axes in the frequency plots.

However, we found cumulative distributions useful. If you have a more specific proposal for replacing cumulative distributions, we would be very grateful to hear it. We also hope that magnifying the figures in TIFF format with a higher resolution will improve visibility.

(4) Figure 2B: This data would be better presented with all regions stretched to the same size (the reason is explained in the public review).

We performed the scaled plots for the stranded SNS-seq origins over the genic and intergenic regions as the reviewer suggested (see Author response image 3), but we prefer to keep the unscaled versions in the manuscript.

**Author response image 3. sa3fig3:** Distribution of mapped origins in scaled genic and intergenic regions. Scaled heatmaps present the distribution of the mapped origins and shuffled controls within scaled genic and intergenic regions (± 2 kb).

(5) Line 149: "The number of origins in both cells was 148 compared using normalised mapped reads": Supplementary Figure 2D mentions that conditions were subsampled to the same amount. I would mention that explicitly in the main text ("compared using normalized, subsampled mapped reads"), as 'normalizing' would not include 'subsampling' for me. Also, I could not find the methods section that the authors refer to here.

Thanks for the suggestion. We changed the text to make this point clearer. In the methods section, the subsampling process was referred to as 'PCF down-sampling', but we changed now the name to 'Read sub-sampling' to be more consistent in the edited version of the manuscript.

(6) Figure 2C: I struggled to understand what gDNA stands for. Maybe it could be replaced with something like distribution in genome?

Thanks for this suggestion. It is changed to ‘distribution in genomic sequence’.

(7) Figure 5C: I cannot see how a G4 30 kb from an origin could be relevant. This also does not fit the scale of the author's own model at all (Figure 8).

The main goal of Figure 5C was to demonstrate the differences between origins and the nearest G4s compared to the shuffled controls. The graph shows that 50% of the origins have a G4 within 2010 bp, whereas the median for the shuffled control is 4154 bp in the case of non-stabilised G4s. Our model is based on Figure 5D, which illustrates the enrichment of G4s and poly(dA) around the centre of origins.

(8) Figure 6B: could be made supplementary in my opinion. All relevant data is repeated in panel D.

It is true that Figures 6B and 6C contain some repetition. However, we would prefer to keep Figure 6B because it provides a quantification of the six indicated categories, along with the statistical tests. Figure 6B only presents the three categories that changed significantly. Figure 6D shows distribution but does not contain quantified data.

(9) Figure 6D: This plot is repeating a lot, within single figures (Figure 6A, top) but also between figures (e.g., Figure 5D, Figure 4B). I'd prefer it if the initial plots of each figure were expanded a bit (here Figure 6A, top) to include some information from the previous figures. Then all these summary plots could be combined into a single figure at the very end (maybe still as different panels to reduce the number of lines in a single plot). Otherwise, each summary plot repeats the tracks of the previous, which becomes very repetitive.

Our model is based on these summary plots, and we calculated the relative distances between the different elements using them. Two elements were repeated in each plot: the positions of poly(dA) and G4s. These two elements served as reference points to determine the relative positions of the other elements. Following your suggestion would result again in repetitive summary plots at the end, as one combined summary plot would be overloaded with lines and difficult to understand.

(10) Figure 6D & Figure 7C: Both show predicted G4s; however, on the plus strand, one prediction has a two-peaked shape, the other only a single peak. Is this a mistake?

The graphs for the predicted G4s do not have the same shape in the two plots as they were performed in different reference genomes for *T. brucei*. Figure 6C is in the 427-reference genome as the MNase-seq data set was analysed in this reference genome and we re-did the SNS-seq analysis and the G4 prediction in this reference genome to be able to compare them directly. In Figure 7C we are comparing origins DRIP-seq and predicted G4s, in this case all datasets could be compared in the 427-2018 reference genome.